# Functional specification of CCK+ interneurons by alternative isoforms of Kv4.3 auxiliary subunits

**Viktor János Oláh[1,2], David Lukacsovich[3], Jochen Winterer[3], Antónia Arszovszki[1], Andrea Lőrincz[4], Zoltan Nusser[4], Csaba Földy[3], János Szabadics[1]\***

[1]Laboratory of Cellular Neuropharmacology, Institute of Experimental Medicine, Budapest, Hungary; [2]János Szentágothai School of Neurosciences, Semmelweis University, Budapest, Hungary; [3]Laboratory of Neural Connectivity, Brain Research Institute, University of Zurich, Zurich, Switzerland; [4]Laboratory of Cellular Neurophysiology, Institute of Experimental Medicine, Budapest, Hungary

**Abstract** CCK-expressing interneurons (CCK+INs) are crucial for controlling hippocampal activity. We found two firing phenotypes of CCK+INs in rat hippocampal CA3 area; either possessing a previously undetected membrane potential-dependent firing or regular firing phenotype, due to different low-voltage-activated potassium currents. These different excitability properties destine the two types for distinct functions, because the former is essentially silenced during realistic 8–15 Hz oscillations. By contrast, the general intrinsic excitability, morphology and gene-profiles of the two types were surprisingly similar. Even the expression of Kv4.3 channels were comparable, despite evidences showing that Kv4.3-mediated currents underlie the distinct firing properties. Instead, the firing phenotypes were correlated with the presence of distinct isoforms of Kv4 auxiliary subunits (KChIP1 vs. KChIP4e and DPP6S). Our results reveal the underlying mechanisms of two previously unknown types of CCK+INs and demonstrate that alternative splicing of few genes, which may be viewed as a minor change in the cells' whole transcriptome, can determine cell-type identity.

**\*For correspondence:**
szabadics.janos@koki.mta.hu

**Competing interests:** The authors declare that no competing interests exist.

## Introduction

The biophysical and morphological properties of complementary GABAergic cell classes are specifically tuned for regulating the activity of the much more populous principal cells in broad temporal (from second to sub-millisecond; *Hu et al., 2014*; *Overstreet-Wadiche and McBain, 2015*) and spatial (from axons to distal dendrites *Freund and Buzsáki, 1996*) domains. Thus, specialized features of GABAergic neurons help the hippocampus to comply with the vast computational demand related to various behaviors (*Klausberger and Somogyi, 2008*). However, the functional diversity of currently known GABAergic cell types cannot match the vast amount of hippocampal behavioral tasks. Therefore, a more complete understanding of GABAergic cells is one of the major goals of current research. Emerging evidence using single-cell transcriptomics suggests that the number of GABAergic types may be higher than previously recognized (*Földy et al., 2016*; *Fuzik et al., 2016*; *Harris et al., 2018*; *Que et al., 2019*; *Tasic, 2018*; *Zeisel et al., 2015*). While these studies accelerated cell classification efforts by identifying a large number of genes that appear to define further subtypes in neuronal taxonomy, the functional relevance of most transcriptomic marker genes remain poorly understood. To better understand the relationship between gene expression and physiological function, and how these properties define neuronal types, we employed a multidisciplinary approach to study a previously unrecognized level of functional diversity among CCK-expressing hippocampal interneurons (CCK+IN) in rat CA3 area.

Several properties distinguish CCK+INs from other major GABAergic cell classes. Unlike in other GABAergic cells, the axons of CCK+INs are highly enriched with CB1 receptors, which mediate activity-dependent regulation of GABA release (*Freund and Katona, 2007*). As a result of this delicate control, CCK+INs are ideally suited for dynamic inhibition of a subset of principal cells based on the context of ongoing activity. The in vitro firing of CCK+INs is believed to be homogeneous typically displaying intermediate AP widths (between that of pyramidal cells and classical fast-spiking interneurons) and clear spike frequency accommodation (*Cea-del Rio et al., 2011*; *Glickfeld and Scanziani, 2006*; *Szabadics and Soltesz, 2009*; *Szabó et al., 2014*). These excitability properties are crucial for the generation of characteristic CCK+INs firing in vivo, which is observed during exploration-associated with theta and gamma oscillations (*Klausberger et al., 2005*; *Lasztóczi et al., 2011*). In contrast to their apparent biophysical homogeneity, the morphology and molecular content of CCK+INs are diverse. Based on their axonal morphology basket-, mossy fiber-associated, Schaffer collateral-associated and perforant path-associated types (*Cope et al., 2002*; *Vida and Frotscher, 2000*; *Vida et al., 1998*) can be distinguished. These distinct morphological types selectively control excitation from different afferents (hence their names). At the molecular level, complimentary expression of marker genes characterize subsets of CCK+INs, such as VGluT3 and VIP (*Somogyi et al., 2004*), and single-cell RNA-sequencing also uncovered several additional molecular varieties (*Fuzik et al., 2016*). Due to their distinction from other major classes and large interclass variability, CCK+INs are ideal for examining relationships between genes and cellular identity and boundaries between GABAergic cell classes.

In this study, we investigated hippocampal CCK+INs from a broader perspective that could reveal previously undetected excitability parameters suitable for specific physiological functions. A hint that such diverse physiological parameters exist came from previous in vivo recordings that showed that individual CCK+INs are differentially active during various oscillatory states (*Klausberger et al., 2005*; *Lasztóczi et al., 2011*), where many of them appeared to prefer either lower or higher frequency ranges. We found two types of hippocampal CCK+INs in rats based on their different excitability, with potentially different contributions to network events, particularly in the range of theta oscillations. Detailed realistic simulations showed that switching only the properties of Kv4.3-mediated currents can sufficiently convert one functional cell type to the other. Combined analyses of the complete mRNA content and protein expression of single cells revealed that the pronounced functional distinction between these two CCK+IN type is defined by differential isoform usage of three auxiliary subunits of the Kv4.3 channels.

## Results

### A large portion of CCK+INs show state-dependent firing in the CA3 area

To explore potential differences in the excitability of individual CCK+INs in the CA3 region of the rat hippocampus, first we characterized their firing properties in two different conditions. Specifically, we recorded their spiking in response to current steps from two, physiologically plausible membrane potential ranges of *post hoc* identified CCK+INs. We focused mostly on the CA3 region because here the diversity of CCK+INs is the largest within the hippocampus. When CCK+INs (n = 557 cells) were stimulated from slightly depolarized membrane potentials (MP, range: $-60 - -65$ mV) relative to rest ($-64.7 \pm 0.4$ mV), action potential (AP) firing always showed spike-frequency accommodation, which is one of the most characteristic features of this cell class (*Cea-del Rio et al., 2011*; *Glickfeld and Scanziani, 2006*; *Szabadics and Soltesz, 2009*; *Szabó et al., 2014*). However, we noticed that numerous CCK+INs (n = 290 cells) showed MP-dependent firing: their initial spiking was strongly inhibited and its onset was delayed when it was evoked from hyperpolarized MPs (between $-75$ to $-85$ mV, *Figure 1A–B*). On average, these cells started firing after a $252 \pm 15$ ms silent period from hyperpolarized MP (measured from the start of the current injection). We named these cells as Transient Outward Rectifying cells or TOR cells (a term that was used to describe cells with similar firing patterns in other brain regions: *Stern and Armstrong, 1996*). The rest of CCK+INs (n = 267 cells) were characterized as regular spiking or RS cells, as they fired regularly irrespective of their MP and they started firing with a short delay ($33 \pm 2$ ms) when stimulated from hyperpolarized

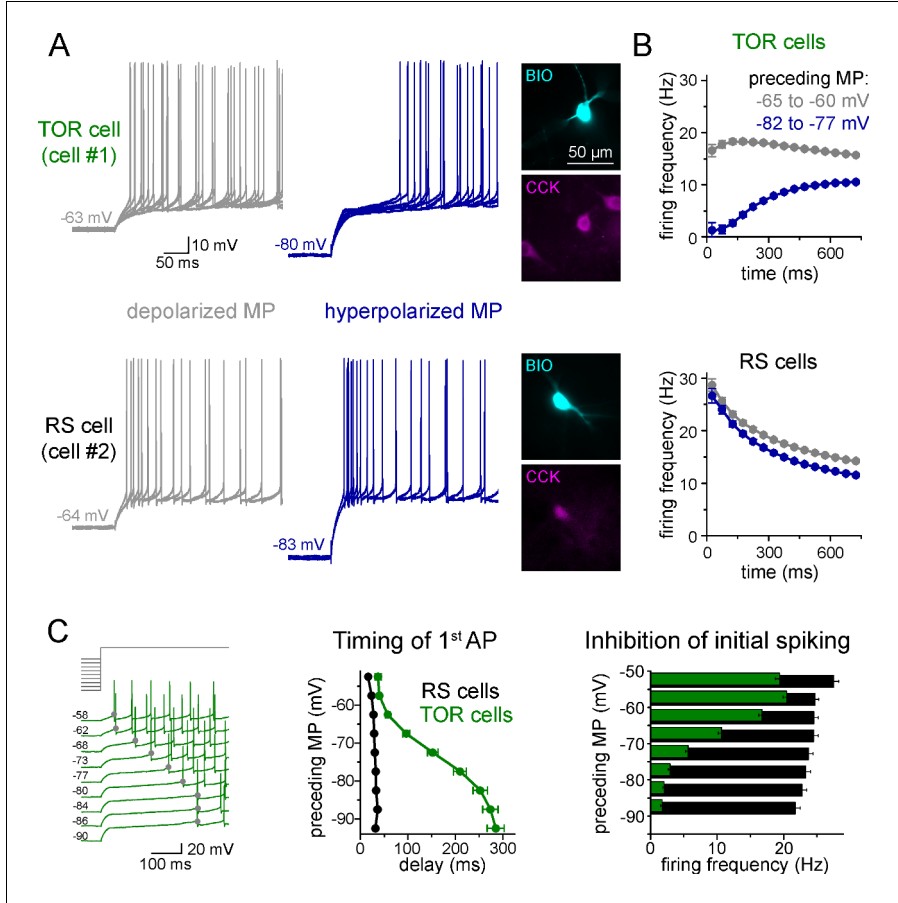

**Figure 1.** Two distinct firing patterns within CA3 CCK+ cells. (**A**) Firing properties of two representative CCK+INs in the CA3 hippocampal region. Firing was elicited with square pulse current injection of identical amplitude, but from depolarized (grey traces), or hyperpolarized MPs (blue traces). Several trials are superimposed to show the stability of the timing of the first action potential. Insets show the immunolabelling of the biocytin filled (BIO) recorded cells for CCK. (**B**) Average time course of AP occurrence in TOR and RS cells from two MP ranges (n = 120 and 113 representative cells, respectively). (**C**) Timing of the first AP and probability of APs during the first 150 ms of the square pulse stimulus shows steep MP-dependence in TOR cells, whereas the initial spikes are stable in the RS cells. The amplitude of stimulating current steps was standardized for each cell and only the preceding holding current (3 s) was varied in individual trials. Traces show a representative recording from a TOR cell. The average data derived from 85 TOR and 81 RS cells.

The online version of this article includes the following source data for figure 1:

**Source data 1.** Electrophysiological parameters of TOR and RS CCK+INs.

MP. At depolarized MP (−55 to −65 mV), the first APs of both TOR and RS cells occurred with similar short delays (48 ± 3 ms and 26 ± 1 ms, respectively, Student t-test, p=0.09, t(160) = −1.706).

Next, we applied a protocol allowing the detailed quantification of the MP-dependence of firing in individual cells (n = 81 RS and 85 TOR cells). Specifically, firing was evoked by a current step that was calibrated for each cell to elicit a similar mean firing frequency (10–20 Hz) from slightly depolarized MP. The holding current preceding the current step was systematically varied to reach a wide range of steady-state MPs (3 s, in the range of −50 and −90 mV, above the theoretical reversal potential of potassium current, *Figure 1C*). In TOR cells, the number of APs within the first 150 ms and the timing of the first AP showed steep voltage dependence ($V_{1/2}$ value of the Boltzmann fits were −67.4 mV and −73 mV, respectively, $R^2$ = 0.995 and 0.999, *Figure 1C*). In contrast, the delay and number of spikes did not show membrane potential-dependence in RS cells. Importantly, both RS and TOR firing was stable in prolonged patch-clamp recordings, which lasted for 45–64 min (average 58 min, n = 4 TOR and 3 RS cells).

In contrast to the difference in initial spiking, other firing and membrane properties of TOR and RS cells were similar, including input resistance, AP threshold, half-width, rate of rise and after-hyper-polarization (*Figure 1—source data 1*). Furthermore, the frequency and amplitude of spontaneous synaptic events (including both IPSCs and EPSCs; 2.47 ± 0.57 Hz and 3.26 ± 0.86 Hz, Student t-test, p=0.45, t(15) = −0.78; −51.6 ± 3.3 pA and −51.5 ± 2.5 pA, Student t-test, p=0.98, t(15) = −0.02, n = 9 TOR and 8 RS cells) were similar. We found TOR cells in the CA1 region as well. However, here their prevalence was lower compared to CA3 (2 TOR out of 13 CA1 CCK+IN). Finally, both TOR (n = 15 cells) and RS (n = 12 cells) CCK+INs were detected in the CA3 region of adult rats (older than 70 days) suggesting that these biophysical features are not age-dependent. In summary, the firing of TOR cells shows a remarkable sensitivity to a physiologically plausible 20 mV shift in the MP, despite having no other distinctive passive electrical or spiking properties compared to RS cells.

## TOR and RS firing types do not correlate with previously known subtypes of CCK+ cells

The CCK+IN class has been previously divided into several subtypes based on various functionally relevant features. Therefore, next we investigated whether the two MP-dependent firing phenotype could be linked to previously identified subtypes of CCK+INs. Three subtypes have been previously identified within CA3 CCK+INs based on the target zones of their axons: (i) basket cells (BCs) that innervate the soma and proximal dendrites of pyramidal cells (*Hendry and Jones, 1985*), (ii) mossy fiber-associated cells (MFA) with axons branching in stratum lucidum and hilus regions that are also occupied by the axons of dentate gyrus granule cells (*Vida and Frotscher, 2000*), and (iii) Schaffer-collateral associated cells (SCA), targeting the same regions as Schaffer collaterals (*Cope et al., 2002*). Altogether, 172 of the recorded CCK+INs were unequivocally identified either as BCs (n = 31), MFAs (n = 96) or SCAs (n = 45; *Figure 2A*). Both TOR and RS firing types occurred similarly among these morphological subtypes (*Figure 2B*) indicating that neither of the two firing phenotypes could be assigned to any of the morphology-based CCK+IN subtypes. This assessment was further supported by quantitative analyses of axonal and dendritic features (*Figure 2—figure supplements 1* and *2*).

In addition to the variable morphological features, CCK+INs heterogeneously express several molecular markers (*Somogyi et al., 2004*). To study their molecular heterogeneity, we performed single cell RNA sequencing (SC-RNAseq) (*Földy et al., 2016*) of individually recorded TOR cells (n = 8) and RS cells (n = 8). Only those cells were analyzed in detail, which had detectable mRNA for both CCK and cannabinoid-receptor type 1 (CB1R or Cnr1; *Figure 2C*). In accordance with previous observations (*Földy et al., 2016*), the included cells contained the transcripts of at least 3000 different genes (the range of detected genes in the included cells was between 4173 and 9629 with mean 6946 ± 430 genes). Most CCK+IN-relevant marker genes (calbindin/Calb1, vesicular glutamate transporter VGlut3/Slc17a8, vasoactive intestinal protein/VIP, serotonin receptor subtype 3a/5HT3a, Reelin, neuropeptide-Y/Npy) were expressed at comparable levels in both TOR and RS cells (*Figure 2C*). One of the exception was the activity-dependent transcription regulator, Satb1 (*Close et al., 2012*), whose mRNA was present in TOR cells (n = 8 out of 8), whereas only 4 of the eight tested RS cells had detectable levels (Mann-Whitney test, p=0.0095, z = 2.59, U = 57). Furthermore, we found that Lypd1, which is enriched in a subpopulation of CCK+INs in the CA1 (*Harris et al., 2018*) was expressed in all tested TOR cells (n = 8) and in only one of the RS cells (1 out of 8, Mann-Whitney test, p=0.0006, z = 3.45, U = 64). Despite these differences, hierarchical cluster analysis of the total transcriptome did not categorize these cells into two distinct groups that would correspond to their firing phenotype (*Figure 2—figure supplement 2*), suggesting that these cells have highly similar transcriptomic identities (*Fuzik et al., 2016*; *Harris et al., 2018*). Our complete SC-RNAseq dataset is available at NCBI GEO as GSE133951.

To confirm these findings at the level of protein expression, we performed immunohistochemistry on identified CCK+ TOR and RS cells in separate experiments (*Figure 2D* and *Figure 2—figure supplement 3*). As expected, CB1R protein was detected in the axon of almost all tested cells (15 out of 18 cells). Furthermore, VGluT3 (23 out of 46 TOR cells, 16 out of 30 RS cells), Reelin (all 11 tested TOR and 7 out 10 RS) and Calb1 (10 out 30 TOR and 7 out of 29 RS) proteins occurred similarly in RS and TOR types. By contrast, while the majority of TOR cells were positive for Satb1 protein (109 out of 132, 82.6%), only few RS cells were Satb1 positive (14 out of 128 tested, 10.9%). Satb1 protein was also prevalent in non-recorded CCK+ cells within the CA3 region of the acute slices (126



**Figure 2.** Different excitability does not correlate with previously known diversity of CCK+ cells. (**A**) Axonal (black) and dendritic (blue) reconstructions of four CCK+INs in the CA3 region representing two major morphological types, the mossy-fiber associated cells and basket cells. The firing of cell#1 and #2 are shown in detail in *Figure 1A*. The firing from hyperpolarized MP is shown for each cell. Immuno-labelling for CCK and the nuclear protein Satb1 are shown next to each recorded cell. (**B**) Prevalence of TOR and RS cells within three major morphological types of CA3 CCK+. Numbers of identified cells are indicated on each bar (BC: CCK+ basket cells, MFA: mossy-fiber associated cells, SCA: Schaffer-collateral associated cells). (**C**) Single cell RNAseq characterization of recorded TOR (left, n = 8) and RS (right, n = 8) cells for known GABAergic cell markers. Each column corresponds to single identified CCK+INs. The bars on the right shows the logarithm of p values of the statistical comparison of TOR and RS cells for each genes. Significantly different mRNA content between TOR and RS cells was found for Satb1 and Lypd1 (p=0.0095 and p=0.0006, Mann-Whitney Test). (**D**) Comparison of immunohistochemical and single cell RNAseq data. Bar plots show the fraction of the recorded cells with detectable RNA content (threshold: 0.2) for the selected markers and immunopositivity for the same proteins (for

*Figure 2 continued on next page*

*Figure 2 continued*

examples of the immunolabelling see *Figure 2—figure supplement 3*). The number of tested cells are shown on each bar.
The online version of this article includes the following figure supplement(s) for figure 2:

**Figure supplement 1.** Quantitative comparison of the dendritic and axonal morphologies of TOR and RS cells.
**Figure supplement 2.** Hierarchical cluster analysis of CCK+INs.
**Figure supplement 3.** Example cells that were immunopositive for CB1R (**A**), Calb1 (**B**), VGluT3 (**C**), Reelin (**D**) and Satb1 (**E**) and the low magnification overview of the labelling pattern of these molecules within the CA3 region.

Satb1+/CCK+ out of 425 CCK+), suggesting that the detection of this activity-dependent protein was not a consequence of patch-clamp recording. The somata of Satb1+/CCK+ cells were found throughout CA3 strata oriens, pyramidale, lucidum and radiatum and their proportion to all CCK+ cells were similar (34.1%, 21.4%, 33.6%, and 27.2%, respectively). In agreement with the lower occurrence of TOR cells in the CA1, non-recorded CCK+ cells in this region were less likely to be Satb1 positive (21.6% of 231 CCK+ cells). In the hilar region of the dentate gyrus the proportion of Satb1 +/CCK+ cells was also low (4.8% of 123 CCK+ cells). We have also detected Satb1 and CCK overlap in tissue samples derived from transcardially perfused animals (20.2%, 97 Satb1+/CCK+ out of 480 CCK+ cells in the CA3 area). To conclude, transcriptomic and immunohistochemical data suggest that Satb1 is predictive of TOR identity in hippocampal CCK+INs and that these TOR and RS firing phenotypes are not related to previously known subtypes within CCK+INs.

## Differences in low-voltage-activated potassium currents (I$_{SA}$) underlie the heterogeneity of CCK+IN firing

Next, we investigated ionic conductances that are responsible for the two firing types in CCK+INs. We focused on near-threshold potassium currents because they have been shown to effectively regulate firing responses under similar experimental conditions in various brain regions (*Lammel et al., 2008*; *Margrie et al., 2001*; *Neuhoff et al., 2002*; *Stern and Armstrong, 1996*). First, we characterized each cell as TOR or RS type under normal recording conditions. Next, we recorded near-threshold potassium currents between −100 and −25 mV, in the presence of voltage-gated sodium channel and hyperpolarization-activated cation channel inhibitors (3 μM TTX and 10 μM ZD7288). *Post hoc* immunolabelling confirmed that all included cells were CCK+. TOR cells had a substantial amount of potassium currents at firing threshold (*Figure 3A*, 683 ± 109 pA at −40 mV, n = 17). By contrast, potassium currents in RS cells activated at more positive voltages and were much smaller at threshold (148 ± 28 pA, n = 11, p=0.0006, t(26)=3.89, Student's t-test). The robust low-voltage-activated potassium currents (I$_{SA}$), which are activated 30 to 40 mV below AP threshold, can underlie the strong inhibition of AP generation in TOR cells, whereas I$_{SA}$ in RS cells is much smaller at near-threshold voltage range.

Next, we investigated the availability of I$_{SA}$ at different MPs. For these measurements, potassium currents were evoked by a voltage step to −30 mV following various pre-pulse potentials between −100 to −35 mV. The V$_{1/2}$ of the average inactivation curve was −64.5 ± 0.2 mV (Boltzmann fit, R$^2$ = 0.999, mean of V$_{1/2}$ from individual cells: −63.8 ± 0.9 mV, n = 13 cells, *Figure 3B*) in TOR cells. Importantly, the majority of I$_{SA}$ was available at slightly hyperpolarized MPs (91.3 ± 1.6% at −80 mV), where the inhibition of firing was clearly observable during the characterization of TOR cells (see *Figure 1* data). But at −60 mV, where the inhibition of spiking was not prominent, the majority of outward currents in TOR cells were inactivated; only 35.7 ± 3.4% of the total current remained available. Similarly to the activation, the inactivation of I$_{SA}$ in RS cells was shifted toward positive voltage ranges (V$_{1/2}$: −57.4 ± 0.3 mV, n = 8 cells, mean of individual data: −55.6 ± 2 mV, comparison with TOR cells, see above, p=0.0006, t(19)=4.12, Student's t-test) and a larger portion of this smaller current was available at −60 mV (52.7 ± 6.3%, *Figure 3B*). Thus, hyperpolarization of the RS cells cannot add a substantial amount of inhibitory conductance.

In addition, the inactivation time constant of I$_{SA}$ was faster in RS cells compared to TOR cells (18.8 ± 1.7 ms, n = 7, vs. 71.2 ± 9.1 ms, n = 14, measured at −25 mV, p=0.0007, t(19)=4.025, Student's t-test). The recovery from inactivation was slower in TOR compared to RS cells, and it showed steep voltage dependence (TOR cells: time constants of the recovery were 58.2 ± 2.9 ms, 46.2 ± 2.9

**Figure 3.** Differences in the Kv4-mediated $I_{SA}$ currents underlie the heterogeneity of CCK+IN firing.
(**A**) Representative traces from two single cells and average voltage dependence of activation of $I_{SA}$ in TOR
(n = 17) and RS (n = 11) cells. The grey dotted line indicates AP threshold ($-37.52 \pm 0.3$ mV) measured before TTX
application. Notice the large amount of outward current in TOR cells at subthreshold MPs. (**B**) Representative
traces and voltage dependence of inactivation of $I_{SA}$ at $-30$ mV from different holding potentials (n = 14 and 10
for TOR and RS cells, respectively). Blue and grey shaded areas indicate the voltage ranges from which state-
dependent firing was tested (see *Figure 1*). (**C**) Representative traces and average voltage dependence of $I_{SA}$
conductance in nucleated patches from TOR and RS cells (n = 7 and 8, respectively). The activation curves are
shown both as the average of raw and normalized data to highlight the differences both in the amount and
voltage dependence of the conductance. Blue points connected in the right panel represent $I_{SA}$ currents in the
presence of HpTX (1 µM). (**D**) Representative CCK+INs firing patterns recorded before, during and after HpTX

*Figure 3 continued on next page*

*Figure 3 continued*

application (in response to identical current steps) and the average AP probabilities in TOR (n = 6) and RS cells (n = 9) from hyperpolarized MP ranges (−82 to −77 mV). (**E**) Average delay of the first AP and reversible effect of HpTX. Connected symbols represent individual measurements (paired t-test: p=0.01). (**F**) HpTx-sensitive $I_{SA}$ in representative TOR and RS cells (left) and the average voltage dependence (n = 7 for both TOR and RS cells). The online version of this article includes the following figure supplement(s) for figure 3:

**Figure supplement 1A.** Representative outside-out patch recording of dendritic $I_{SA}$ current from a RS cell, which was somatically loaded with Alexa594 dye to visualize dendrites in epifluorescent illumination and to record firing pattern.

**Figure supplement 2.** TOR phenomenon is not affected when Kv1 and Kv3 channels are inhibited.

ms, 29.8 ± 2.5 ms and 4.4 ± 0.1 ms at −65,–75, −85 and −120 mV, respectively; whereas in RS cells: 11.8 ± 4.4 ms, 8.0 ± 2.2 ms, 7.1 ± 1.6 ms and 3.3 ± 0.4 ms, respectively). Due to these properties, $I_{SA}$ currents in RS cells remained limited around AP threshold either from −80 or −60 mV preceding voltage. In summary, the different properties of $I_{SA}$ currents, particularly the left-shifted inactivation and activation curves, may underlie the differences in TOR and RS firing phenotypes.

In our whole-cell recordings the majority of the membrane conductances remained intact, which prevented fully controlled voltage clamping at more depolarized MPs and the determining the exact voltage dependence of $I_{SA}$. Therefore, we compared the $I_{SA}$ in TOR and RS cells in nucleated patch recordings. The better voltage control allowed us to record currents at more depolarized voltage ranges and compare the total density of $I_{SA}$ in the two types of CCK+INs. The density of $I_{SA}$ in TOR cells was smaller compared to RS cells (*Figure 3C*; TOR: 1.91 ± 0.46 nS/pF, RS: 3.86 ± 0.8 nS/pF, n = 7 and 8). However, consistent with our findings using whole-cell configuration, $I_{SA}$ activated at a more hyperpolarized voltage range in TOR cell nucleated patches ($V_{1/2}$ TOR: −16.4 ± 4.7 mV, $V_{1/2}$ RS: −8.9 ± 1.7 mV, with 7% and 1.4% median, 15 ± 6.5% and 3.6 ± 1.8% average activation at −40 mV, respectively, Mann-Whitney test, p=0.024, z = 2.257, U = 48, n = 7 and 8) and the current had slower inactivation time constant in TOR (38.1 ± 4.7 ms, n = 6) compared to RS (16.8 ± 5 ms, n = 8, p=0.011, t(12) = 2.984) cells. Thus, in spite of the smaller conductance density, $I_{SA}$ generated larger inhibitory charge in TOR cells than in RS cells during 100 ms long voltage steps to −40 mV (0.47 ± 0.1 and 0.21 ± 0.05 pC/pF, p=0.032, t(13) = 2.394, n = 7 and 8).

Because whole-cell and nucleated patch configurations do not report on the subcellular distribution of $I_{SA}$ currents, we investigated local $I_{SA}$ currents by pulling outside-out patches from the soma and dendrites of RS and TOR cells (*Figure 3—figure supplement 1*). These recordings confirmed that the density of somatic $I_{SA}$ was significantly larger in RS cells (70.5 ± 15.2 pA/µm$^2$, n = 26) compared to that of the TOR (42.2 ± 5.2 pA/µm$^2$, n = 17, p=0.047, t(41) = −2.05) cells. The density of dendritic $I_{SA}$ was similar in TOR and RS cells (19.6 ± 3.3 pA/µm$^2$, n = 53 vs. 19.5 ± 3.0 pA/µm$^2$, n = 47). Furthermore, we did not observe a significant gradient along the dendrites (up to 300 µm from soma; TOR: R$^2$ = 0.011, p=0.21, n = 56; RS: R$^2$ = 0.01, p=0.22, n = 53). On average, the kinetics of patch currents matched those of whole-cell currents and confirmed slower inactivation in TOR cells. However, there was a pronounced variability of $I_{SA}$ between individual patches from TOR, but not RS, cells (*Figure 3—figure supplement 1D*), even when multiple patches were pulled from the same cell. In many TOR cell patches the current kinetics resembled the patch- and whole-cell currents of the RS cells. While in other TOR cell patches much slower currents were also detected. This variability might be due to the clustered occurrence of voltage-gated potassium channels in GABAergic cells (e.g. see *Kollo et al., 2006*).

## Kv4 channels are responsible for both types of $I_{SA}$ currents in CCK+INs

Next, we investigated the identity of potassium channel subunits responsible for the differences of $I_{SA}$ in RS and TOR cells and for the MP-dependent firing in TOR cells. Neither TEA (0.5 and 10 mM, blocking Kv3 channels) nor a low concentration of 4-AP (100 µM, blocking Kv1 and Kv3 channels) eliminated the initial firing gap in TOR cells (*Figure 3—figure supplement 2A*). Only a high concentration of 4-AP (5 mM) was able to diminish the TOR phenomenon. These, together with the above determined low voltage activation properties, suggest the involvement of Kv4 channels (*Lien et al., 2002*), of which Kv4.3 has been shown to be present in hippocampal CCK+INs (*Bourdeau et al., 2007*; *Kollo et al., 2006*). To specifically test the contribution of Kv4 channels to the TOR

phenomenon, we applied Heteropodatoxin-1 (HpTX, 1 µM), which selectively inhibits Kv4.2 and Kv4.3 subunit-containing channels by shifting the voltage dependence of the activation (*DeSimone et al., 2011*; *Sanguinetti et al., 1997*). In the presence of HpTX, significantly more APs were evoked in TOR cells during the first 125 ms of the stimulus compared to control conditions in the same cells (*Figure 3D*, from −80 mV preceding MP, 9.7 ± 2.9% vs 5.1 ± 2.2%, n = 9, p=0.0037, t (5) = −5.12, paired t-test). Furthermore, the delay of the first APs was reversibly shortened from 239 ± 61 ms to 116 ± 38 ms (*Figure 3E*, p=0.01, t(7) = 3.493, paired t-test, n = 8 TOR cells). However, HpTX did not change the number and temporal distribution of APs in TOR cells from −60 mV (data not shown) and the firing of RS cells also remained unchanged both at −80 and −60 mV (n = 8 cells). Furthermore, HpTX had only minor effects on the half-width of the APs (TOR cells: control: 0.48 ± 0.02 ms, HpTX: 0.52 ± 0.03 ms, n = 8, p=0.15, t(7) = −1.6; RS cells: control: 0.55 ± 0.04 ms, HpTX: 0.59 ± 0.06 ms, n = 8, p=0.11, t(7) = −1.83). Finally, the HpTX effect was completely reversible on the TOR phenomenon, which provides an additional evidence for the stability of the TOR phenomenon in individual cells during long recording periods (25–58 min). In summary, HpTX-sensitive, Kv4-mediated currents are crucial for the TOR phenomenon.

Next, we analyzed the contribution of HpTX-sensitive currents to $I_{SA}$ in TOR and RS cells. In whole-cell conditions, both types had substantial amounts of HpTX-sensitive currents. In TOR cells, the HpTX-sensitive currents activated at more negative voltage (*Figure 3F*) and the time constant of inactivation was slower than in RS cells (49.1 ± 12.6 ms vs. 11.8 ± 1.6 ms, at −25 mV). The voltage dependence of inactivation of the HpTX-sensitive current was also left-shifted in TOR relative to RS cells ($V_{1/2}$ TOR: −64.0 ± 0.6 mV, n = 7; RS: −56.8 ± 1.3 mV, n = 7). In nucleated patch experiments, consistent with the known gating modification effects of HpTX (*DeSimone et al., 2011*; *Sanguinetti et al., 1997*), the activation voltage of $I_{SA}$ was shifted to more depolarized membrane potentials in both TOR and RS cells (change in $V_{1/2}$ TOR: 12.4 ± 3.3 mV, n = 4; RS: 21.5 ± 6.9 mV, n = 5; *Figure 3C*), which resulted in a similar reduction in the conductance at −30 mV (TOR: 60 ± 10%, n = 4; RS: 61 ± 16%, n = 5). To conclude, our results suggest that the majority of inactivating $I_{SA}$ potassium currents are mediated by Kv4 channels in both TOR and RS cells, in spite of the different properties of the currents.

## Realistic models of TOR and RS firing

In the previous experiments, we examined excitability properties in CCK+INs using steady-state current injections. However, excitation of neurons in vivo is more dynamic and the membrane potential constantly fluctuates according to the ongoing state of the network. The different temporal properties of $I_{SA}$ in TOR and RS suggest that these cells may differently follow membrane potential fluctuations. Therefore, to predict the frequency ranges of oscillations that are optimal for the distinct CCK+IN types with distinct excitability and the brain states that TOR and RS cells favors, we simulated the realistic dynamic behavior of the two phenotypes.

First, we equipped the morphology of 10 reconstructed CCK+INs (5 TOR and 5 RS cells) with known voltage-dependent conductances and passive properties of hippocampal CCK+INs (*Bezaire et al., 2016*; *Figure 4—source data 1*) and reproduced core CCK+IN firing (*Figure 4A*). Next we used two sets of $I_{SA}$ conductances based on our outside-out patch recordings and tuned their densities to recreate RS and TOR firing properties in simulations. In the models, $I_{SA}$ and firing properties were best reproduced if RS cells were equipped with a single type of $I_{SA}$ potassium conductance ($I_{SA}$RS). Consistent with our recordings, $I_{SA}$ currents of TOR cells were reproduced by a mixture of two $I_{SA}$: $I_{SA}$RS and a left-shifted, slowly inactivating current ($I_{SA}$TOR) in a 3:1 ratio (*Figure 4B*). After adding $I_{SA}$RS or $I_{SA}$TOR+RS to the core CCK+IN properties (*Bezaire et al., 2016*) in current-clamp simulations, the models reproduced the common firing properties of CCK+INs (including AP width, peak and frequency and AHP shape), but in the presence of $I_{SA}$TOR+RS all 10 cells showed MP-dependent firing with similar temporal dynamics and voltage dependence as recorded experimentally. When the same cells were equipped with $I_{SA}$RS only, they turned to RS firing type (*Figure 4—figure supplement 1*). Thus, swapping $I_{SA}$RS to $I_{SA}$TOR+RS alone is sufficient to generate TOR properties, even if the reconstructed cell originally belonged to the RS type and vice versa. Importantly, the simulations confirmed that TOR firing required less $I_{SA}$ potassium conductance, than RS firing in the same reconstructed cells (57.9% less; TOR: 1.72 ± 0.27 mS/cm$^2$, RS: 2.97 ± 0.47 mS/cm$^2$).



**Figure 4.** Different $I_{SA}$ currents in TOR and RS cells tune them for different network states. (**A**) $I_{SA}$TOR+RS and $I_{SA}$RS were added to 10 reconstructed CCK+INs, which possess every known voltage-dependent conductance (***Bezaire et al., 2016***). Changing $I_{SA}$RS to $I_{SA}$TOR+RS in the same cells transformed the firing from RS to TOR phenotype. These simulated cells were equipped with synaptic conductance to simulate input drives in various network states. (**B**) Voltage clamp simulations with complete morphology and realistic conductances reproduced whole-cell $I_{SA}$ currents of TOR and RS cells, including voltage-dependence, the total current measured at the soma (left and middle graphs), and inactivation and the kinetics of inactivation and recovery from inactivation (right graph, where recovery was measured at −120,–85, −75 and −65 mV). (**C**) Representative traces showing the number and temporal distribution of 8 Hz-modulated synaptic inputs to simulated CCK+INs and its MP and $I_{SA}$ conductance in two conditions with either $I_{SA}$RS or $I_{SA}$TOR+RS. (**D**) Average effects of exchanging $I_{SA}$RS to $I_{SA}$TOR

*Figure 4 continued on next page*

*Figure 4 continued*

+RS on the output of 10 CCK+INs during various input frequency ranges (x-axis) and baseline output activity (i.e. with I$_{SA}$RS conductance, y-axis). Yellow color shows small change in firing when I$_{SA}$TOR+RS replaced I$_{SA}$RS, whereas red color indicates robust reduction in AP output. Representative traces on the right depict three examples with different input frequencies. Red triangles highlight inhibited spikes.

The online version of this article includes the following source data and figure supplement(s) for figure 4:

**Source data 1.** Average conductance densities and membrane properties of ten simulated cells.
**Figure supplement 1.** Realistic CCK+INs in-silico.

## TOR cells are selectively silenced by I$_{SA}$TOR in a narrow range of oscillatory states

Next, we investigated the activity of RS and TOR firing cells during in vivo-like simulations. For this, we modeled excitatory and inhibitory inputs arriving onto the somato-dendritic axis of the 10 reconstructed CCK+INs. The occurrence of excitatory events was clustered and tuned to frequencies ranging from 1 to 100 Hz, to mimic oscillations, with kinetic parameters derived from the above experiments (see Materials and methods). All 10 model cells received the same input patterns and we varied excitatory strength as a parameter (by changing the number of EPSCs, *Figure 4C*). This configuration recapitulated oscillating membrane potentials (from −72 mV to threshold) and also a wide range of spiking frequencies in CCK+INs representing the frequency ranges that have been observed in vivo (*Klausberger et al., 2005*; *Lasztóczi et al., 2011*).

Next, we compared the average spiking of I$_{SA}$TOR+RS potassium conductance-equipped cells (n = 10 cells) with the spiking of the same cells during the same conditions except that they were equipped with I$_{SA}$RS only. We found that CCK+INs were efficiently silenced by I$_{SA}$TOR+RS, as compared to I$_{SA}$RS, in the 8–15 Hz input frequency range. On average, 37.2 ± 0.4% fewer APs were evoked (see red areas in the middle of the *Figure 4D* graph and example traces, quantified in the 4–10 Hz output range). By contrast, during lower and higher input frequencies the presence of I$_{SA}$TOR+RS only slightly reduced firing (spiking was decreased by 9.8 ± 0.3% and 8.8 ± 0.2%, between 1–6 Hz and 25–100 Hz, respectively). To summarize, I$_{SA}$TOR+RS conductance alone enables CCK+INs to be selectively silenced during 8–15 Hz input regime. This single conductance changed the way TOR and RS cells integrate and respond to physiologically relevant input patterns. This observation adds a novel level of complexity to the diverse functions of GABAergic cells and can contribute to their observed heterogeneous firing during different network states (*Klausberger et al., 2005*; *Klausberger and Somogyi, 2008*; *Lasztóczi et al., 2011*). The input frequency dependence of the inhibition of firing can be explained by the specific temporal properties of I$_{SA}$TOR+RS. Specifically, in addition to the voltage dependence of activation and inactivation (*Figure 3*), the time constant of inactivation (*Figure 3—figure supplement 1D*) and the recovery from inactivation determines the difference in availability of these currents during various oscillatory states. Thus, minor modifications in the properties of I$_{SA}$ enable distinct functions in individual cells.

## Molecular identity of Kv4.3 channels in TOR and RS cells

Next, we aimed to uncover molecular differences in TOR and RS cells that could explain their different I$_{SA}$ properties. First, we checked the expression of Kv4 channel coding genes in RS and TOR cells using the SC-RNAseq data (*Figure 5A–B*). Kv4.2 was absent in most of the tested 16 CCK+INs in accordance with previous findings (*Bourdeau et al., 2007*; *Rhodes et al., 2004*). The mRNA of Kv4.3 subunits was detected in most CCK+INs (15 out of 16 cells), including both RS and TOR types. This was consistent with the finding that both RS and TOR cells possess HpTX-sensitive currents. Interestingly, in line with the predictions of the simulation, the average expression of Kv4.3 was not lower in RS than in TOR cells (RS: 2.03 ± 0.4 versus TOR: 1.56 ± 0.34, Mann-Whitney, p=0.37, z = −0.89). We found differences in other Kv channels as well, such as Kv3.2, Kv1.3 and Kv1.6 subunits (*Figure 5A*). However, because these subunits do not generate low-voltage-activated, inactivating currents, and/or they are sensitive to low concentrations of 4-AP and TEA (*Figure 3—figure supplement 2*), we did not investigate further the currents generated by these subunits (*Lien et al., 2002*). To confirm these transcriptomic findings, we performed immunohistochemistry to localize the Kv4.3 subunits. Overall, we observed an intense neuropil labelling for Kv4.3 in DG, CA3 strata

**Figure 5.** Similar Kv4.3-expression and different auxiliary subunit, KChIP and DPP-content in CCK+INs.
(**A**) Normalized gene count of primary and auxiliary subunits of voltage-gated potassium channels from single cell RNAseq data of TOR (n = 8) and RS (n = 8) cells. (**B**) Percentage of recorded cells with detectable levels of Kv4.3 mRNA (left bars, n = 8 and 9 cells) and protein (right bars, n = 20 and 23 tested cells). (**C–D**) Immunofluorescent co-localization of CCK and Kv4.3 in a TOR (**C**), and a RS cell (**D**) in CA3 stratum lucidum. (**E**) Percentage of recorded cells with detectable levels of KChIP1 mRNA (n = 8 and 8 cells) and proteins (n = 20 and 27 cells). (**F–H**) Immunofluorescent co-localization of CCK with CB1 (green) and KChIP1 in a TOR (**G**), and a RS cell (**H**) from the same slice shown in low magnification image (**F**). (**I**) Percentage of recorded cells with detectable levels of KChIP4 mRNA (left bars, n = 8 and 8 cells) and protein (right bars, n = 29 and 30 tested cells). (**J**) Major KChIP4 splicing

*Figure 5 continued on next page*

*Figure 5 continued*

isoforms consist of different exons in the N-terminal region (represented as red boxes). Each row represents a single cell in the color-mapped data and columns correspond to individual exons aligned to the schematic illustration of isoforms above. Red and blue colors code high and low mRNA levels, respectively. The average exon counts from the two types of CCK+INs (n = 8 and 8) are shown at the bottom using the same color code scheme. (K) Percentage of recorded cells with detectable levels of DDP6 and DPP10 mRNA. (L) Assembly of major DPP6 isoforms and exon levels in individual CCK+INs are shown as above (J).

The online version of this article includes the following figure supplement(s) for figure 5:

**Figure supplement 1.** Kv4.3 and KChIP immunohistochemistry in perfusion or immersion fixed CA3 slices.

radiatum and oriens, but not in the CA1 area. A subset of INs were also labelled throughout the hippocampus (*Bourdeau et al., 2007*; *Rhodes et al., 2004*). Kv4.3 proteins appeared to be enriched in the somatic and dendritic plasma membranes, but the subcellular distribution was often uneven and clustered (*Figure 5D* and *Figure 5—figure supplement 1A*) in line with the known distribution pattern of the Kv4.3 subunit (*Kollo et al., 2006*). Next we analyzed Kv4.3 immunolabelling in biocytin-filled CCK+INs. In agreement with our pharmacology and SC-RNAseq data, the Kv4.3 signal was present in both TOR (*Figure 5B-D*, 15 out of 20 tested cells) and RS CCK+INs (19 out of 23 cells). However, the immunosignal was usually stronger in RS than in TOR cells even within the same section, which was consistent with our observations from nucleated patch recordings (*Figure 3*). The same trend was observable in the CA3 area obtained from perfusion-fixed brain (*Figure 5—figure supplement 1A*), in which the Kv4.3 signal was detectable at strong or moderate level in most CCK+/Satb1- cells (putative RS cells, 99%, 338 out of 342 tested cells) in contrast to weak or hardly detectable labelling in most CCK+/Satb1+ cells (putative TOR cells, 97%, 78 out of 80 tested cells). In summary, immunolabelling confirms that TOR cells have a lower density of Kv4.3 channels, which is critical in their unique firing.

## Auxiliary subunits of Kv4 channels in TOR and RS cells

Core Kv4 channel proteins form ternary complexes with dipeptidyl aminopeptidase-like proteins (DPLPs, including DPP6 or DPP10) and K$^+$ channel interacting proteins (KChIP1-4 from Kcnip1-4 genes), which fundamentally influence current properties. These auxiliary subunits are expressed in various splicing isoforms (*Jerng and Pfaffinger, 2014*; *Pongs and Schwarz, 2010*), and their large diversity allows delicate modification of Kv4-mediated current properties. For example, most KChIP isoforms promote surface expression and accelerate inactivation and recovery from inactivation. However, the so-called transmembrane KChIPs (tmKChIPs) have opposite effects as they can retain Kv4 from the plasma membrane and decelerate inactivation (*Holmqvist et al., 2002*; *Jerng and Pfaffinger, 2008*; *Jerng and Pfaffinger, 2014*; *Pruunsild and Timmusk, 2012*).

KChIP1, as one of the classical KChIPs, accelerates inactivation and increases surface expression of Kv4.3 channels (*Beck et al., 2002*; *Bourdeau et al., 2011*; *Jerng and Pfaffinger, 2014*; *Pongs and Schwarz, 2010*). We detected KChIP1 mRNA in most RS cells (6 of 8 tested), but it was absent in 5 out of 8 TOR cells. The average mRNA level was significantly higher in RS than in TOR cells (RS: 2.27 ± 0.5, n = 8; TOR: 0.87 ± 0.42, n = 8; Mann-Whitney, p=0.025, z = −2.25). KChIP1 immunohistochemistry revealed an even clearer distinction between RS and TOR cells (*Figure 5E*). KChIP1 protein was detected in the majority of RS cells (20 out of 27), whereas only few TOR cells showed positive immunoreaction (5 out of 21). Furthermore, the signal in the positive TOR cells was typically weaker compared to RS cells (*Figure 5F–H*). We also found that the KChIP1 signal was present not only in the plasma membrane, but also in the cytosol, in agreement with their role in trafficking (*Pongs and Schwarz, 2010*). This data from recorded cells were confirmed by the analysis of CCK+INs in perfusion fixed brains (*Figure 5—figure supplement 1A*). We detected strong KChIP1 signal in the majority of CCK+/Satb1- cells (corresponding to RS cells, 122 out of 132 tested cells, 92.4%). In contrast, only 4.9% of CCK+/Satb1+ cells (corresponding to TOR cells, 2 out of 41 tested cells) showed strong KChIP1 immunosignal. With regards to other KChIPs, while we did not detect significant amounts of KChIP2 and KChIP3 mRNAs, the splicing-invariant sequence of KChIP4s was detected in all TOR cells (8 out of 8), but only in 3 out of 8 RS cells. In biocytin-labelled cells, the available antibody detected KChIP4 protein only in a very few CCK+INs regardless of their firing

type (4 out of 29 TOR, 5 out of 30 RS cells, *Figure 5I* and *Figure 5—figure supplement 1B–E*), which was in apparent contradiction with our RNAseq data. However, we can explain this contradiction based on the following observations. In general, the relatively weak KChIP4 immunosignal in CCK+INs was surrounded by strong neuropil labelling in the stratum radiatum (*Figure 5—figure supplement 1B*). While KChIP1 is expressed only in INs, KChIP4 is known to be associated with Kv4.2 channels in pyramidal cells as well (*Rhodes et al., 2004*). At high magnification (*Figure 5—figure supplement 1D*), we found KChIP4 signal to be associated with tube-like structures, likely representing the plasma membranes of putative pyramidal cell dendrites. However, our KChIP4-specific antibody was raised against a long amino acid segment, which includes the highly variable N-terminal region, which endow various KChIP4-isoforms with different effects on Kv4 channel function (*Holmqvist et al., 2002*; *Jerng and Pfaffinger, 2008*; *Jerng and Pfaffinger, 2014*; *Pruunsild and Timmusk, 2012*). This clue prompted us to further analyze the mRNA data at the level of individual KChIP4 exons (*Figure 5J*). KChIP4e belongs to the so-called tmKChIPs (*Jerng and Pfaffinger, 2008*), which, in contrast to most KChIP types and isoforms, do not promote the plasma membrane expression of Kv4 and have opposite effects on the inactivation kinetics (*Jerng and Pfaffinger, 2008*; *Pruunsild and Timmusk, 2012*). We detected the KChIP4e isoform-specific exon in 7 TOR cells (out of 8 tested). KChIP4b, which acts as classical KChIPs, was detected in only one TOR cell (from eight tested). Out of the 3 RS cells that had detectable KChIP4 levels, one expressed KChIP4e specific exons, whereas other two expressed the KChIP4b isoform. Altogether, these data indicate that RS cells express KChIP1, whereas TOR cells primarily express KChIP4e. We suggest that the differential expression of these isoforms underlie some of the distinct properties of $I_{SA}$ in RS and TOR cells because KChIP1 is known to accelerate inactivation and increase surface expression, whereas tmKChIPs, such as KChIP4e, do not facilitate Kv4 plasma membrane trafficking and confer slow channel inactivation.

However, KChIPs alone cannot account for all differences between $I_{SA}$ currents in TOR and RS cells as they cooperate with DPLPs to fine-tune Kv4 functions. DPP10 and DPP6 proteins effectively shift the voltage dependence of Kv channel activation and inactivation (*Jerng et al., 2007*; *Jerng and Pfaffinger, 2012*; *Jerng and Pfaffinger, 2014*; *Nadal et al., 2006*; *Nadal et al., 2003*; *Pongs and Schwarz, 2010*). Therefore, next we analyzed the expression of these molecules in the RNAseq data (*Figure 5K–L*). Conserved sequences of DPP10 were present in both TOR (7 out of 8) and RS (8 out of 8) cells and the DPP10c was the dominant isoform in all cells regardless of their firing type. DPP6 gene was also detected in most cells regardless of their firing type. At isoform level, RS cells primarily expressed DPP6L. By contrast, TOR cells expressed a combination of DPP6S (7 out of 8 cells) and/or DPP6L-specific exons (4 out of 8 cells). To conclude, in addition to KChIPs, DPLP isoform expression displays correlation with TOR and RS firing types of CCK+INs.

## Discussion

### $I_{SA}$ current properties tune CA3 CCK+IN function

CCK+INs can dynamically control the activity of selected cell assemblies (*Freund and Katona, 2007*). However, during network oscillations, the activity of individual CCK+INs is highly variable and individual CCK+INs display different preferences for distinct oscillatory regimes (*Klausberger et al., 2005*; *Klausberger and Somogyi, 2008*; *Lasztóczi et al., 2011*). Our results describe two novel CCK+IN populations, TOR and RS cells in the CA3, and provide a mechanistic explanation for their activity patterns during 8–15 Hz theta oscillations. We found that under basal conditions TOR cells are unlikely to contribute to theta oscillations unless they are primed by preceding depolarizations. However, during lower and faster network oscillations, there may be no difference in TOR and RS cell functions. Because distinct network state-dependent activity is considered as a cell type classification criteria (*Klausberger and Somogyi, 2008*), TOR and RS cells can be viewed as separate partitions within the CCK+IN class. The TOR phenomenon potentially explains the variable recruitment of individual CCK+INs during subsequent theta cycles enabling state and activity history-dependent control of network functions. As a key feature of CCK+INs, effective neurotransmitter release from these cells often require sustained activity (*Földy et al., 2006*; *Losonczy et al., 2004*). Thus, the elimination of multiple AP-consisting bursts in TOR cells during the theta range (*Figure 4D*) is

expected to largely diminish the GABAergic inhibition conveyed by this specific subset of CCK+INs onto CA3 neurons.

Despite the functional distinction between TOR and RS cells, their morphology, basic electrophysiological characteristics and overall transcriptomic profiles are surprisingly similar. Genes and proteins were present in both types similarly that have been shown previously to delineate certain subpopulations within CCK+INs (such as VGluT3 or Calb1). Thus, TOR and RS cells cannot be distinguished based on their general gene expression profiles, which is recently used for identification of individual cell types (*Fuzik et al., 2016*; *Harris et al., 2018*). Furthermore, RS and TOR cells are also morphologically diverse, targeting either or both somata and dendrites with their axons. Thus, TOR and RS types cannot be assigned with the previously established morphological types of CCK+INs. Biophysical signatures of AP shapes, intrinsic excitability and synaptic inputs are also similar in RS and TOR cells. Thus, TOR and RS distinction of CCK+INs apparently does not comply with morphological and genomic cell type-definitions. Instead, the functional divergence of TOR and RS types is allowed by the modification of a single ionic conductance. Kv4.3 current alone is sufficient to switch between TOR and RS firing modes. Changing only $I_{SA}$RS to $I_{SA}$TOR+RS in the same realistically modelled CCK+INs specifically silenced them during 8–15 Hz input regimes and reproduced the recorded differences in their firing. Pharmacology confirmed the crucial role of Kv4.3 channels in TOR firing. However, RS cells also express Kv4.3 channels, and in fact, their density is higher in RS cells compared to TOR cells. Thus, paradoxically, in spite of Kv4.3-mediated currents are responsible for the unique and distinctive firing properties of TOR cells, Kv4.3 currents are present in both cell types. We found that a potential explanation for this paradox is the differential expression of auxiliary subunits of Kv4.3 channels (see below).

The transcription regulating function of Satb1 depend on activity (*Close et al., 2012*) and the function of KChIPs is under the control of activity-related intracellular $Ca^{2+}$ signaling. This raises the possibility that RS and TOR firing modes are convertible in CCK+INs depending on neuronal activity. However, we did not detect changes in the firing patterns – including the length of self-inhibition in TOR cells – during long recording sessions indicating that under our experimental conditions CCK+INs stably maintain RS or TOR firing.

## Same channel protein, but distinct auxiliary subunits may be responsible for different $I_{SA}$ currents in TOR and RS cells

Although fine-tuned combination of largely different conductance sets can converge to similar firing patterns (*Marder and Goaillard, 2006*), a single conductance can completely convert the way a neuron elicits APs. The majority of fundamental spiking properties were similar in TOR and RS cells. However, due to unique properties of Kv4.3-mediated current, TOR cells generate late firing – similarly as in other brain regions (*Stern and Armstrong, 1996*; *Zheng et al., 2019*) – and are unresponsive to theta frequency band inputs. While the Kv4.3 channel was present both in RS and TOR cell types, the availability of auxiliary subunits was different. KChIP1 is strongly expressed by RS cells, whereas most TOR cells lacked this cytosolic auxiliary subunit. The known effects of the KChIP1 (*Beck et al., 2002*; *Bourdeau et al., 2011*; *Jerng and Pfaffinger, 2014*; *Pongs and Schwarz, 2010*) correlate well with the properties of the $I_{SA}$ in RS cells. As a classical KChIP, KChIP1 increases surface expression of Kv4 channel complex. Indeed, we measured larger density of $I_{SA}$ current in outside out and nucleated patches from RS than in TOR cells. We also observed that Kv4.3 immunosignal was usually stronger in RS compared to TOR cells. Furthermore, we were able to reproduce TOR and RS firing phenotypes in the realistically simulated conditions only if the total amount of $I_{SA}$ conductance was larger in RS than in TOR cells. The apparent paradox between larger current density and the smaller measured current amplitude near the AP threshold in RS cells derives from the differences in the voltage dependence of activation in RS and TOR cells. $I_{SA}$ channels are submaximally activated at physiological subthreshold voltage. However, because of the left-shifted activation curve, a much larger fraction of $I_{SA}$ channels is activated in TOR cells at lower voltage ranges. Therefore, larger currents can be generated even by a smaller number of channels. Another substantial influence of KChIP1 on Kv4.3 is the acceleration of steady-state inactivation kinetics and the recovery time from inactivation (*Beck et al., 2002*; *Bourdeau et al., 2011*; *Jerng and Pfaffinger, 2014*; *Pongs and Schwarz, 2010*). These are also correlated well with the differences of $I_{SA}$ in RS and TOR cells, as both parameters were much faster in the former type of cells.

KChIP proteins and their splice-variants show unusual functional diversity. Different splicing of the same protein can have opposing effects on Kv4 functions, whereas, splice-variants of different proteins can have analogous effects. One outstanding group is the tmKChIP family that consist of KChIP2x, KChIP3x, KChIP4a and KChIP4e. Their common structural feature is an extra N-terminal hydrophobic domain that binds them to the membrane (*Holmqvist et al., 2002*; *Jerng and Pfaffinger, 2008*; *Jerng and Pfaffinger, 2014*; *Pongs and Schwarz, 2010*). In contrast to classical KChIPs, tmKChIPs typically retain Kv4 channels from the plasma membrane, slow the inactivation kinetics and the recovery from inactivation. KChIP4 was the dominant KChIP in TOR cells and it was not detected in most RS cells. TOR cells expressed a tmKChIP isoform, KChIP4e. The slower inactivation and recovery of $I_{SA}$ in KChIP4e-expressing TOR cells were critical for their unique functionality during 8–15 Hz network states because they define the availability of the $I_{SA}$ conductance. KChIP4 was present only in three RS cells and two of them expressed the KChIP4b isoform, which is a classical KChIP (*Jerng and Pfaffinger, 2008*). In contrast to classical KChIPs, all tmKChIPs including KChIP4e do not facilitate surface expression of Kv4 channels (*Holmqvist et al., 2002*; *Jerng and Pfaffinger, 2008*; *Pruunsild and Timmusk, 2012*). As explained above, several lines of evidence suggest lower Kv4.3 channel density in the KChIP4e-expressing TOR compared to KChIP1-expressing RS cells. The exact effects of KChIP4e on the kinetics of Kv4.3-mediated currents are not known. Albeit little data is available about the modification of Kv4 gating properties by KChIP4e in expression systems, its structural similarities suggest that it acts like other better studied tmKChIPs. The presence of other tmKChIPs results in slow inactivation kinetics, often slower than that of the solitary Kv4 channels (*Holmqvist et al., 2002*; *Jerng et al., 2007*; *Jerng and Pfaffinger, 2008*; *Tang et al., 2014*). As a consequence of enhanced closed-state inactivation of Kv4.3 channels, KChIP4a causes a leftward shift in the voltage dependence of inactivation (*Tang et al., 2013*; *Tang et al., 2014*). Albeit it is a likely possibility due to the similarity of their N-terminal domains, it remains to be answered whether the effects of KChIP4e on Kv4.3 kinetics are similar to the other studied tmKChIPs. Altogether, the presence of KChIP4e in TOR cells and KChIP1 in RS cells is consistent with their different $I_{SA}$ kinetics and densities that underlie the different functionality of these cells. Future studies are expected not only to confirm that these alternatively spliced variants are solely responsible for the two firing types, but can also address whether these differences in subunits are determined by the destiny of the cells from early of their development or whether these subunits are actively regulated throughout the life span and may underlie activity-dependent modification of the CCK+INs population.

We should also note that the apparent discrepancy in KChIP4 detection between the mRNA and protein levels may also be explained by the presence of different isoforms, as our antibody may have targeted the highly variable N-terminal region.

Both DPLPs were present in TOR and RS cells. DPP10c isoform, which is known to affect voltage dependence of Kv4 channels, but does not accelerate inactivation (*Jerng et al., 2007*), was ubiquitous in both types of CCK+INs. Albeit all tested CCK+INs had a significant amount of DPP6 mRNA, TOR and RS cells expressed different isoforms. RS cells expressed only DPP6L and in TOR cells the primary isoform was DPP6S (but DPP6L was also present in several cells). DPP6S can contribute to the left-shifted voltage dependence of activation of $I_{SA}$ in TOR cells. The left-shifted voltage dependence is important for the sufficient prevention of spiking. The dual set of DPP6 proteins fits well with our observations that many properties of TOR cells can be described only if two populations of Kv4.3-mediated currents are present. Altogether, these observations suggest that apparently small modifications in the available components of ion channel complexes underlie the different functions of TOR and RS cells.

The effects of KChIPs are not isolated from the other auxiliary subunits of Kv4. The various stoichiometries of individual Kv4 channels with DPPs and KChIPs allow delicate settings of the channel kinetics. The net effects of KChIPs and DPLPS are not simply the sum of the effects of individual subunits, and the combinatorial possibilities are not yet fully explored with the known 17 variants of KChIPs and 8 variants of DPPs. Some subunits can dominate others or the interaction can result in unexpected effects (*Jerng et al., 2005*; *Nadal et al., 2006*; *Zhou et al., 2015*). For example, when DPP6K is present, KChIP4e causes a leftward shift of the voltage dependence of inactivation and deceleration of the recovery from inactivation compared to other KChIPs (*Jerng and Pfaffinger, 2012*). In this regard, it is an important observation for our results that DPP6S, unlike some other DPLPs, cannot overcome the tmKChIP4-mediated deceleration of inactivation (*Jerng et al., 2007*; *Jerng and Pfaffinger, 2008*; *Seikel and Trimmer, 2009*). Thus, the expression of DPP6S in TOR

cells may preserve the unique modulatory effects of KChIP4e. Because of the composition of these ternary channel complexes by multiple proteins and splicing variants, whose interactions are not yet characterized, the exact contribution of individual components of DPP6S/L/DPP10c-KChIP4e-Kv4.3- and DPP6L/DPP10c-KChIP1-Kv4.3-complexes cannot be predicted yet. In addition to the direct modulation, Kv4 protein complexes can be phosphorylated by various kinases and are involved in complex post-phosphorylation signaling (*Hu et al., 2020*), which require the presence of auxiliary subunits and modify the mediated currents (*Vacher and Trimmer, 2011*). Thus, in spite of the large number of potential mechanisms that can modulate Kv4.3 functions, all differences that we observed between RS and TOR cells (i.e., the higher channel density, faster inactivation kinetics and faster recovery from inactivation of $I_{SA}$ in RS compared to TOR cells) are consistent with the differential expression of KChIP1 and KChIP4e subunits. In combination with the cell type-specific expression and contributions of DPP10c, DPP6L and DPP6S, these results may explain the unique voltage-dependency in the two types of CCK+INs without the involvement of additional differences between RS and TOR cells. Thus, the different firing properties and responsiveness during 8–15 Hz network states of RS and TOR cells can be established by surprisingly minor modifications in a few auxiliary subunits.

## Materials and methods

Animal protocols and husbandry practices were approved by the Institute of Experimental Medicine Protection of Research Subjects Committee (MÁB-7/2016 for slice recording and anatomy experiments and MÁB-2/2017 for immunolabelling experiments in perfusion fixed brains) and by the Veterinary Office of Zurich Kanton (single cell RNAseq experiments).

### Slice preparation, solutions and chemicals

Hippocampal slices were prepared from 21 to 33 days old Wistar rats (deeply anaesthetized with isoflurane) in ice-cold artificial cerebrospinal fluid (85 mM NaCl, 75 mM sucrose, 2.5 mM KCl, 25 mM glucose, 1.25 mM NaH$_2$PO$_4$, 4 mM MgCl$_2$, 0.5 mM CaCl$_2$, and 24 mM NaHCO$_3$, Leica VT1200 vibratome). The slices were incubated at 32°C for 60 min after sectioning and were then stored at room temperature until they were used. The normal recording solution was composed of 126 mM NaCl, 2.5 mM KCl, 26 mM NaHCO$_3$, 2 mM CaCl$_2$, 2 mM MgCl$_2$, 1.25 mM NaH$_2$PO$_4$, and 10 mM glucose. For standard recordings, pipettes were filled with an intracellular solution containing 90 mM K-gluconate, 43.5 mM KCl, 1.8 mM NaCl, 1.7 mM MgCl$_2$, 0.05 mM EGTA, 10 mM HEPES, 2 mM Mg-ATP, 0.4 mM Na$_2$-GTP, 10 mM phosphocreatine, and 8 mM biocytin (pH 7.2; 270–290 mOsm). Chemicals for the intra- and extracellular solutions were purchased from Sigma-Aldrich, ion channel blockers were from Tocris or Alomone and fluorophores were from Invitrogen.

### Somatic recordings

For recordings, cells in slices were visualized with an upright microscope (Eclipse FN-1; Nikon) with infrared (900 nm) Nomarksi DIC optics (Nikon 40x NIR Apo N2 NA0.8W objective). Electrophysiological recordings were performed at 34.5–36°C. During current-clamp recordings, firing patterns were elicited by square shaped current pulses with increasing amplitudes (starting from −100 pA up to 700 pA, Δ20 pA, duration: 1 s) or with standard steps (eliciting average firing between 10–20 Hz), which was preceded by 3-second-long different amplitude holding current steps (−450 to 120 pA, with 20 or 30 pA increments) to reach preceding MP range between −90 and −50 mV. We aimed to restrict hyperpolarization above the reversal of potassium ions, which may case shifts in the ionic equilibrium and affect the excitability. However, we did not observe such effects even in those cases when the membrane potential was below the reversal of potassium. AP distributions were calculated from all recorded traces, which contained action potentials and binned by 50 ms from each recorded cell. In these recordings, the pipette capacitance was neutralized (2.5–5 pF remaining capacitance) and bridge balance compensation was set to eliminate apparent voltage offsets upon current steps. AP threshold was determined as the voltage corresponding to 50 mV/ms slope. Voltage values were not corrected for the liquid junction potential (theoretically: −15.4 mV). Traces were low pass filtered at 6–20 kHz and digitized at 40–100 kHz using a Multiclamp 700B amplifier with Digidata 1440 A interface (Molecular Devices).

For voltage-clamp recordings, cells were patched in normal extracellular solution to first record their firing patterns, and then 2.5 μM TTX and 10 μM ZD7288 was added to reduce sodium and Ih currents respectively. Voltage protocols for potassium current measurements consisted of a 300 ms long conditioning pulse at −120 mV, followed by a 300 ms long voltage steps between −120 and −20 mV (for current activation and decay time constant measurements), and a last voltage step to −30 mV for 100 ms (to measure voltage dependence of potassium current inactivation). Series resistances were between 6–20 MΩ (75–80% compensated with 53 μs lag) and were constantly monitored. Data were discarded if the series resistance changed more than 25%.

The recovery from inactivation test protocol consisted four voltage steps: (1) −120 mV for 345 ms, allowing full recovery, (2) −30 mV for 500 ms, resulting complete inactivation, (3) −65, −75,−85 or −120 mV steps with variable duration between 1–233 ms, (4) −30 mV test pulse. The area of current (which is less sensitive to filtering than the peak) was analyzed according to the voltage and duration of the preceding recovery step. To isolate I$_{SA}$ currents from non-inactivating current control traces (steps 1 and 3 were set to −50 mV with identical duration) were subtracted from each trace. Furthermore, these measurements were performed in the presence of TTX, 1 mM 4-AP and 10 mM TEA to reduce contamination from non-Kv4 channels.

## Outside-out patch recordings

To obtain outside-out patch recordings, first we made somatic whole-cell recordings from the selected cells using IR-DIC optics. This was necessary for classifying their firing as TOR or RS cells, for loading biocytin for subsequent CCK immunolabelling, and for loading 20 μM Alexa-594 fluorescent dye, which allowed the visualization of their dendrites. After at least a 5-minute-long loading period the somatic pipette was retracted. Intact dendrites (30–80 μm from the surface) were visualized and patched using epifluorescent illumination (less than a minute illumination time). The thick-wall pipettes (resistance: 20–55 MΩ) were filled with 5 μM Alexa-594. After break-in, we confirmed that no neighboring structure was loaded with the fluorescent dye and outside-out configuration was achieved by the slow retraction of the pipette. By applying similar voltage steps as described for whole-cell voltage-clamp configurations, we recorded dendritic potassium currents (without pharmacological isolation). After outside-out patch recordings, the pipette was pushed into a Sylgard ball (Sylgard 184, Merck), and capacitive responses to voltage steps were recorded and subtracted from the capacitive responses recorded in outside-out patch configuration. The surface area of the patch membrane was calculated as described earlier (*Gentet et al., 2000*) with the specific membrane capacitance determined from 18 nucleated patch experiments (cm = 1.015 ± 0.014 μF/cm$^2$). Distances of the recording sites from the soma were measured based on posthoc epifluorescent or confocal images. Somatic outside-out patches were obtained using the same protocol, with similar pipettes. Current traces were low pass filtered at 10 kHz and digitized at 100 kHz. Capacitive membrane responses were digitized at 250 kHz, without filtering. Leak and capacitive current components were subtracted during potassium current recordings using online p/−4 method. Inactivating potassium currents (at 0 mV) were isolated offline by subtracting currents obtained with a −50 mV prepulse from currents measured with a prepulse of −80 mV.

## Nucleated patch recordings

First we made whole-cell recordings from the selected cells to determine their TOR or RS identity. Then, TTX (1 μM), ZD7288 (10 μM) and TEA (10 mM) were added to the normal extracellular solution, to block sodium currents, Ih currents and a fraction of potassium currents, respectively. TEA does not affect Kv4 channels (*Lien et al., 2002*). Negative pressure was applied through the pipette to bring the nucleus close to the pipette tip, which was then slowly retracted and the plasma membrane sealed around the nucleus. These nucleated patches from somatic membrane were voltage clamped at a holding potential of −80 mV. Voltage commands were identical as in whole cell voltage protocols, except the test voltages ranged between −100 mV and 70 mV allowed by the better voltage control of these isolated membranes. In some of the experiments, when the recording was stable, HpTx (1 μM) was applied from a nearby (<100 μm) small diameter (2–4 μm) pipette to block Kv4 mediated currents. Peak potassium currents were converted to conductance by normalization to the driving force (step voltage minus −90 mV). The capacitance of the excised membrane was determined by dividing the weighted double-exponential fits of the transient responses after the onset of

a −40 mV voltage step in a 5 ms time window with the input resistance of the recorded structure. Conductance density was determined by dividing the obtained conductance value with the capacitance of the structure. Series resistances were between 5–15 MΩ (86–98% compensated with 53 µs lag). Leak and capacitive current components were subtracted during potassium current recordings using online p/−4 method. Traces were low pass filtered at 10 kHz and digitized at 100 kHz. Because the removal of the nucleus damaged the cell body, the identity of these cells was determined with CB1 immuno-labelling in their preserved axons.

Note that the three different recording configurations (whole-cell, outside-out and nucleated patch) may be affected by different conditions, such as local shifts on the ionic milieu, space clamp issue or junction potentials. Therefore, the absolute voltage and kinetics values are not necessarily comparable between the different recording conditions. However, currents from TOR and RS were compared only under the same conditions, thus, their differences were relevant and also consistent in the three recording configurations.

## Anatomical and immunohistochemical characterization of CCK+ cells

All recorded cells were filled with biocytin and processed for immunhistochemistry. After the recordings, slices were fixed in 0.1 M phosphate buffer containing 2% paraformaldehyde and 0.1% picric acid at 4°C overnight. After fixation, slices were re-sectioned at 60 µm thickness (*Neubrandt et al., 2018*). Immunopositivity for CCK was tested with a primary antibody raised against cholecystokinin (1:1000, CCK, Sigma-Aldrich, Cat# C2581, RRID:AB_258806, rabbit polyclonal). The general labelling with this antibody was similar as with a previously used antibody against pro-CCK, including a double neuropil band labelling in the dentate moleculare, perisomatic staining of scattered interneurons throughout the hippocampus and strong axonal bleb labelling at the surface of acute slices (*Szabadics and Soltesz, 2009*). Biocytin labeling was visualized with either Alexa 350-, Alexa 488-, Alexa 594- or Alexa 647-conjugated streptavidin. For additional neurochemical markers, further immunolabeling was used either against Vesicular Glutamate Transporter 3 (1:2000, VGluT3; Merck, Cat#AB-5421, RRID:AB_2187832; guinea pig polyclonal), Cannabinoid Receptor type 1 (1:1000, CB1R, Cayman Cat#10006590, RRID:AB_10098690, rabbit polyclonal), special AT-rich sequence-binding protein-1 (1:400, Satb1; Santa-cruz Cat#sc-5989, RRID:AB_2184337, goat polyclonal, 1:400), Reelin (1:400, Merck Cat#MAB5364, RRID:AB_2179313, mouse monoclonal) or calbindin (1:1000, Calb1; Swant Cat#300, RRID:AB_10000347, mouse monoclonal). Only those cells were included in the analysis, which were positive for CCK (or for CB1, n = 15 cells).

Known morphological subtypes of CCK+INs in the hippocampal CA3 region were determined based on the layer-preference of their axonal arborization. Schaffer collateral associated cells (SCA) (*Cope et al., 2002*) innervate dendrites in the stratum radiatum. Albeit the definition of basket cells (BCs) (*Hendry and Jones, 1985*) and mossy-fiber associated cells (MFA) (*Vida and Frotscher, 2000*) are clear, with termination zones in the strata pyramidale and lucidum, respectively, their practical identification is more complicated because both cell types have a substantial amount of axons in the adjacent strata, especially near to their soma. Therefore, only cells with extended axonal arbor (at least reaching 200 µm from soma) were identified either as BC or as MFA. In this distal region, BCs had clearly targeted stratum pyramidale, whereas distal axons of MFA ran in the stratum lucidum parallel with the cell layer, and they often invaded the hilar region of the dentate gyrus.

## Analysis of potassium channel subunits using immunohistochemistry

For perfusion fixed brain samples two (P25 and P45) Wistar rats were anesthetized and perfused through the aorta with 4% paraformaldehyde and 15 v/v% picric acid in 0.1M Na-phosphate buffer (PB, pH = 7.3) for 15 min. Immunohistochemistry was performed on 70 µm thick free floating coronal sections from the hippocampus. We used a different fixation protocol for acute hippocampal slices containing biocytin-filled cells. After short recordings (5–15 min), slices were transferred to a fixative containing 2% paraformaldehyde and 15 v/v% picric acid in 0.1 M PB for 2 hr at room temperature. Slices were washed in PB, embedded in agarose and re-sectioned to 70–100 µm thick sections. Sections were blocked in 10% normal goat serum (NGS) in 0.5 M Tris buffered saline (TBS) for one hour and incubated in a mixture of primary antibodies for overnight at RT. The following antibodies were used: rabbit polyclonal anti-CCK antibody (1:500, Sigma-Aldrich Cat# C2581, RRID:AB_258806) mixed either with a mouse monoclonal anti-KChiP1 (1:500, IgG1, UC Davis/NIH NeuroMab Facility

Cat# 75–003, RRID:AB_10673162), a mouse monoclonal anti-KChiP4 (1:400, IgG2a, UC Davis/NIH NeuroMab Facility Cat# 75–406, RRID:AB_2493100), or a mouse monoclonal anti-Kv4.3 antibody (1:500, IgG1, UC Davis/NIH NeuroMab Facility Cat# 75–017, RRID:AB_2131966). In some sections, a rabbit polyclonal anti-CB1 receptor (1:2000, Cayman Chemical Cat# *10006590, RRID*:AB_10098690) was also used together with the anti-CCK antibody to more reliably characterize filled cells. Potential TOR cells were identified by Satb1 immunolabelling in the perfusion fixed sections. Satb1 and Kv4.3 were visualized by the same secondary antibody but their labelling pattern could be reliably distinguished based on their different subcellular locations. The following secondary antibodies were used to visualize the immunoreaction: Alexa 488-conjugated goat anti-rabbit (1:500, Thermo Fisher Scientific), Cy5 conjugated goat anti-rabbit (1:500, Jackson ImmunoResearch), Alexa 488-conjugated goat anti-mouse IgG1 (1:500, Jackson ImmunoResearch), and Cy3 conjugated goat anti-mouse IgG1 or IgG2a (1:500, Jackson ImmunoResearch) IgG-subclass-specific secondary antibodies. Biocytin was visualized with Cy5 conjugated Streptavidin (1:1000, Jackson ImmunoResearch). High magnification fluorescent images were acquired with an Olympus FV1000 confocal microscope using a 60x objective (NA = 1. 35).

## Single-cell mRNA analysis

Single-cell mRNA was performed using the Clontech's SMARTer v4 Ultra Low Input RNA Kit. Cells were collected via pipette aspiration into sample collection buffer, were spun briefly, and were snap frozen on dry ice. Samples were stored at −80℃ until further processing, which was performed according to the manufacturer's protocol. Library preparation was performed using Nextera XT DNA Sample Preparation Kit (Illumina) as described in the protocol. Then, cells were pooled and sequenced in an Illumina NextSeq500 instrument using $2 \times 75$ paired end reads on a NextSeq high-output kit (Illumina). After de-multiplexing the raw reads to single-cell datasets, we used Trimmomatic and flexbar to remove short reads, remove adapter sequences and trim poor reads. The remaining reads were aligned to the GRCm38 genome with STAR aligner. Aligned reads were converted to gene count using RSeQC. All data analyses were performed using python3. The analysis included the removal of poor quality cells (at least 3000 genes were detected in each cell), normalization of gene expression data using scran, and analysis of differential expression of genes across cell types. Sequences of splicing variants of KChIP4, DPP6, and DPP10 were validated in the NCBI Genebank and UCSC databases.

The inclusion criteria to the analysis were the expression both Cck and Cnr1. Furthermore, to exclude cells that were potentially contaminated by other cells we considered the following genes as microglia- (Tyrobp, Ctss, C1qc, Cyba, Ly86), astrocyte- (Gfap, Aldh1l1, Sox9, S100b, Slc1a2), oligo-dendrocyte- (Olig1, Fgf2, Mtcp1, Olig2, Olig3) and pyramidal cell-specific (Baiap2l2, Slc17a7, Ptk2b, Nrn1, Fhl2, Itpka, Neurod6, Nptx1, Sv2b, Kcnv1) (*Luo et al., 2019*), and tested these against our single-cell samples. We found that one cell, which met the above criteria, showed significant expression of all five microglia-related genes. This single cell was excluded from the study. Genes related to apoptosis (Bcl2, Casp2, Casp8, Fas; not shown) were absent or present in low copy numbers in both RS and TOR cells indicating that the two firing phenotypes are not due to differential damage during slice preparation. The incomplete biocytin-labelling of most RNAseq analyzed cells prevented us to perform detailed analysis of their morphology. Nevertheless, these sample included at least one SCA, MFA and BC cells.

## Computational modelling

We performed computer simulations using the NEURON simulation environment (version 7.5 and 7.6, downloaded from http://neuron.yale.edu). To create a realistic model of CCK+INs, first we made detailed reconstructions of ten biocytin labelled, electrophysiologically characterized cells (using Neurolucida and the Vaa3D software *Peng et al., 2010*. The passive electrical parameters of the simulated cells were set as follows: first, axial resistance values were set to 120 Ωcm, then specific membrane capacitance and leak conductances (the substrate of membrane resistance) were fitted based on passive membrane responses to small amplitude (20 pA) current injections (ranges: cm: 1.05–0.82 μF/cm$^2$, gl: $5*10^{-5}$ – $5.8*10^{-5}$ S/cm$^2$). The set of active conductances were selected based on a previous publication (*Bezaire et al., 2016*). Conductance densities were adjusted for each individual cell to reproduce firing characteristics representing our average measurements.

Additionally, two variants of A-type potassium conductance models were created. First, the ratio of somatic and dendritic inactivating potassium conductances was set to 2.96:1 somatic to dendritic ratio (see *Figure 4—source data 1*). Based on the observation, that patches pulled from RS cells produced inactivating potassium currents with similar MP-dependence to those recorded in whole-cell configuration, we modelled a single potassium conductance ($I_{SA}$RS) constrained on somatic whole-cell recordings (HpTx-1 sensitive currents measured in RS cells, *Figure 3F*), with the appropriate uncompensated series resistance implemented in the models (2.76 ± 0.28 MΩ, n = 14). For TOR cell models we used a mixture $I_{SA}$ consisting of $I_{SA}$RS and $I_{SA}$TOR, to account for the variability of patch currents from TOR cells. $I_{SA}$TOR has a left shifted voltage dependence and slower inactivation than $I_{SA}$RS. $I_{SA}$TOR+RS reproduced potassium currents from our whole-cell measurements and the typical characteristics of TOR firing. Thus, each of the ten model neuron was simulated both as an RS and TOR phenotype by changing the ratio of gTOR and gRS conductances. Specifically, RS firing type was achieved by adding gRS with $297*10^{-5} \pm 47.2*10^{-5}$ S/cm$^2$ density to the soma and $99.1*10^{-5} \pm 15.7*10^{-5}$ S/cm$^2$ to the dendrites. Whereas, when TOR phenotype was generated with the same active conductance set in the same reconstructed cells, gTOR was added at $44*10^{-5} \pm 7*10^{-5}$ S/cm$^2$ to the soma and $14.7*10^{-5} \pm 2.33*10^{-5}$ S/cm$^2$ to the dendrites and gRS was reduced to $128*10^{-5} \pm 20.3*10^{-5}$S/cm$^2$ at the soma and $42.7*10^{-5} \pm 6.78*10^{-5}$ S/cm$^2$ in the dendrites (*Figure 4—figure supplement 1*). Thus, the overall density of $I_{SA}$ was larger when the cell was in the RS firing mode. Control simulations showed that combined gRS and gTOR better represents TOR firing than a larger amount of gTOR alone.

To investigate the behavior of these model cells in in vivo relevant conditions constructed physiologically plausible input conditions with a large number of temporally organized synaptic excitation and inhibition. For these, first we measured the amplitude and kinetics of glutamatergic and GABAergic currents in TOR and RS cells, using intracellular solutions, containing CsCl (133.5 mM CsCl, 1.8 mM NaCl, 1.7 mM MgCl2, 0.05 mM EGTA, 10 mM HEPES, 2 mM Mg-ATP, 0.4 mM Na2-GTP, 10 mM phosphocreatine, and 8 mM biocytin, pH: 7.2; 270–290 mOsm). To investigate isolated excitatory or inhibitory events, 5 µM SR 95531 hydrobromide (6-Imino-3-(4-methoxyphenyl)−1(6H)-pyridazinebutanoic acid hydrobromide) or 10 µM CNQX (6-Cyano-7-nitroquinoxaline-2,3-dione) and 20 µM D-AP5 (D-(-)−2-Amino-5-phosphonopentanoic acid) was added, respectively. At the end of these recordings, the identity of the recorded synaptic events was confirmed by the application of the above mentioned specific antagonists. These recordings showed no significant difference in the amplitude of excitatory events in TOR and RS cells (TOR: −43.4 ± 0.6 pA, RS: −41.6 ± 0.4 pA, *Mann-Whitney test; p=0.1398, U = 3.42002*106, Z = 1.47654*). The simulated excitatory conductances corresponded to these events as their magnitude followed a normal distribution (mean: 0.22 nS, variance: 0.01 nS). The simulated inhibitory conductance represented both tonic and phasic inhibition (model distribution mean: 2 nS, variance: 0.1 nS) and was based on the similar inhibitory events recorded in CCK+INs (TOR: −89.1 ± 1.4 pA, RS: −78.1 ± 1.7 pA). Synaptic conductances were distributed along the dendrites and somatas of the simulated cells uniformly. Physiologically plausible membrane potential oscillations were driven by these synaptic inputs at different frequency ranges. Specifically to group inputs into a specific frequency, excitatory events were aggregated into normal distributions packages at various frequencies to recreating in vivo relevant MP oscillations in single cells, as follows:

$$onset_{Glut} = \frac{1000}{freq} * 0.633 + \left(69.25112 * \left(0.67747^{freq}\right)\right) + \frac{1000}{freq} \tag{1}$$

where *onset_{Glut/italic}* is the timing of an individual excitatory event, and *freq* is the frequency by which excitatory packages occur. This equation is necessary for setting the width of each normal distribution according to the desired frequency, and therefore keeping the sinusoid shape of the MP during the simulated oscillations. Inhibitory synaptic inputs prevented over-excitation and they followed a uniform random temporal distribution. In these simulations, 20 oscillatory cycles or in case of high frequencies at least 5 s simulation times were used. Before each run, excitatory and inhibitory event amplitudes and onsets were randomized in a unique but reproducible manner (pseudo-randomization with seed value). If the RS model ($I_{SA}$RS) elicited APs, simulations were repeated with TOR model ($I_{SA}$TOR+RS). The amount of excitation was set to produce firing frequencies below 50 Hz in RS cells. Changes in firing rate caused by the replacement of $I_{SA}$RS with $I_{SA}$TOR+RS was calculated

by subtraction of the TOR firing rate from RS firing rate and normalized to the latter. Simulations were run on the Neuroscience Gateway (*Sivagnanam et al., 2015*).

## Data analysis and statistics

Data was analysed using Molecular Devices pClamp, OriginLab Origin, Microsoft Excel software and Python-based scripts. Normality of the data was analyzed with Shapiro-Wilks test. Data are presented as mean ± s.e.m.

## Acknowledgements

We thank Professors Henry Jerng, Paul Pfaffinger and Tõnis Timmusk for advices on DPP6 and KChIP isoforms and Kenneth Harris for suggestions about the gene profiles of hippocampal CCK+INs. We are thankful for the computational resources provided by the Neuroscience Gateway. We thank Andrea Szabó, Dóra Rónaszéki, Dóra Kókay and Andrea Juszel for their their excellent technical assistance and László Barna for the kindly provided microscopy support at the Nikon Microscopy Center at the IEM HAS, which is sponsored by Nikon Europe, Nikon Austria and Auro-Science Consulting. This work was supported by Wellcome Trust International Senior Research Fellowship 087497, Hungarian Brain Research Program KTIA_13_NAP-A-I/5, European Research Council Consolidator Grant (ERC-CoG 772452) to JS, Stephen W Kuffler Research Foundation to VJO, Swiss National Science Foundation (CRETP3_166815 and 31003A_170085) and from the Dr. Eric Slack-Gyr Foundation (Switzerland) to CF. ZN is the recipient of a European Research Council Advanced Grant (ERC-AdG 787157), and a Hungarian National Brain Research Program (NAP2.0) grant.

## Additional information

### Funding

| Funder | Grant reference number | Author |
| --- | --- | --- |
| Wellcome Trust International Senior Research Fellowship | 087497 | János Szabadics |
| Hungarian National Brain Research Program | KTIA_13_NAP-A-I/15 | János Szabadics |
| European Research Council Consolidator Grant | ERC-CoG 772452 | János Szabadics |
| Stephen W. Kuffler Research Foundation | | Viktor János Oláh |
| Swiss National Science Foundation | CRETP3_166815 | Csaba Földy |
| Swiss National Science Foundation | 31003A_170085 | Csaba Földy |
| Dr. Eric Slack-Gyr Foundation (Switzerland) | | Csaba Földy |
| European Research Council Advance Grant | ERC-AdG 787157 | Zoltan Nusser |
| Hungarian National Brain Research Program Grant (NAP2.0) | | Zoltan Nusser |

The funders had no role in study design, data collection and interpretation, or the decision to submit the work for publication.

### Author contributions

Viktor János Oláh, Conceptualization, Formal analysis, Investigation, Visualization, Writing - review and editing; David Lukacsovich, Antónia Arszovszki, Formal analysis, Investigation; Jochen Winterer, Investigation, Writing - review and editing; Andrea Lőrincz, Conceptualization, Formal analysis, Investigation, Writing - review and editing; Zoltan Nusser, Csaba Földy, Conceptualization, Funding

acquisition, Writing - review and editing; János Szabadics, Conceptualization, Formal analysis, Funding acquisition, Writing - original draft

### Author ORCIDs
Zoltan Nusser ![ORCID] https://orcid.org/0000-0001-7004-4111
János Szabadics ![ORCID] https://orcid.org/0000-0002-4968-2562

### Ethics
Animal experimentation: Animal protocols and husbandry practices were approved by the Institute of Experimental Medicine Protection of Research Subjects Committee (MÁB-7/2016 for slice recording and anatomy experiments and MÁB-2/2017 for immunolabelling experiments in perfusion fixed brains) and by the Veterinary Office of Zurich Kanton (single cell RNAseq experiments).

### Decision letter and Author response
Decision letter https://doi.org/10.7554/eLife.58515.sa1
Author response https://doi.org/10.7554/eLife.58515.sa2

## Additional files
### Supplementary files
• Transparent reporting form

### Data availability
Sequencing data have been deposited in GEO under accession code GSE133951.

The following dataset was generated:

| Author(s) | Year | Dataset title | Dataset URL | Database and Identifier |
|---|---|---|---|---|
| Oláh VJ, Lukacsovich D, Winterer J, Lörincz A, Nusser Z, Földy C, Szabadics J | 2019 | Functional specification of CCK+ interneurons by alternative isoforms of Kv4.3 auxiliary subunits | https://www.ncbi.nlm.nih.gov/geo/query/acc.cgi?acc=GSE133951 | NCBI Gene Expression Omnibus, GSE133951 |

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
