## [Decision Letter]

**Acceptance summary:**

The authors demonstrate the existence of functional subpopulations of CCK+ interneurons based on their firing pattern. They identified the underlying mechanism of this diversity and show the potential physiological relevance of the distinct firing patterns in relation to oscillations. The study uses comprehensive and complementary approaches, the quality of the data is high, and the analysis is thorough and carefully executed. The central question is important, that CCK neurons play an important role in information processing in the hippocampal network.

**Decision letter after peer review:**

[Editors’ note: the authors submitted for reconsideration following the decision after peer review. What follows is the decision letter after the first round of review.]

Thank you for submitting your work entitled "Functional specification of CCK+ interneurons by alternative isoforms of Kv4.3 auxiliary subunits" for consideration by *eLife*. Your article has been reviewed by three peer reviewers, one of whom is a member of our Board of Reviewing Editors, and the evaluation has been overseen by a Senior Editor.

Thank you for submitting the manuscript to *eLife*. After careful revision by three reviewers, we concluded, that the study is very interesting. However, based on the long list of comments we are afraid that the revision will take a substantial period of time and therefore needed to reject the manuscript. We would, however, be interested in re-reviewing the study once the authors performed a thorough and in-depth revision of the main concerns listed below and re-submitted their work to *eLife*. Of course, at this time you are free to submit elsewhere. If you don't agree with these reviews and the requested new work, we assume you will take the work elsewhere. If you agree that the manuscript would benefit from the additional work then you can resubmit it to us, with a detailed description of the new data. This is in keeping with our unwillingness to ask authors to do experiments that they don't agree will benefit their manuscript.

Essential revisions:

1) Improve the cell identity following the suggestions of reviewer #2.

2) Perform additional controls for patch seq analysis as suggested by reviewer #2.

3) Improve the discussion on the potential functional role of both interneuron types as questioned by reviewers #1 and 2.

4) Discuss the similarities and differences between native and recombinant channels KChIP4e. If data to recordings from recombinant channels are available, include them in the study. If they are not available, discuss the limitations of your data (see comment of reviewer #3).

5) Improve the discussion on errors related to voltage-clamp recordings of K^+^ channel-mediated currents.

6) Activation and inactivation curves of several ion channels are affected by patch configuration and time-dependent changes, e.g. shifts to more negative values over time (Donnan effects may contribute to these changes). This raises the question whether the curves were similar in different recording configurations and affected by nonstationarities. At the very least, statements should be added to the Materials and methods section.

7) Discus the opposing views that different K^+^ channels or fine-tuning in gating of the same K^+^ channels determines the action potential phenotype as suggested by reviewer #3.

8) Improve the discussion on the correlative nature of evidences between alternatively spliced auxiliary subunits and their effects on neuronal activity patterns.

9) Improve the modeling section by following the suggestions of reviewer #3.

10) Discuss, that acute modifications such as phosphorylation could also influence the gating kinetics of the channels.

11) Increase the number of modeled single cells following reviewer #1.

Reviewer #1:

This study examines the diversity of CCK-expressing interneurons in CA3. Electrophysiological recordings show that CA3-CCK cells can be distinguished in two types based on their firing dynamics into (a) regular spiking (RS) and (b) transient outward rectifying (TOR) interneurons. Interestingly, morphological and intrinsic physiological properties are similar between the two groups of CCK cells. Electrophysiological and pharmacological analysis indicates that Kv4.3 conductances are important in shaping the discharge pattern of TOR cells. However, the study shows that RS cells also express this channel type. Indeed, the density of Kv4.3 is even higher in RS than TOR interneurons. The authors provide the explanation that auxiliary subunits, which seem to differ between the two CCK types, and which can modify the gating properties of Kv4.3 channels, may explain the differential gating characteristics of the channel and the discharge patterns among the two cell types. One of the auxiliary subunits is the KChip protein. KChip1 is strongly expressed in RS cells and only rarely in TOR neurons. In contrast, KChip4 was largely found in TOR but not in RS cells. Further differential expression profiles were observed for DPLPs. RS cells express only DPP6L and TORs largely DPP6S. The main question is, whether minor changes in auxiliary subunit compositions indeed may underlie the significant differences in discharge patterns observed between TOR and RS cells.

1) In view of the fact that a large body of modelling experiments are based on only three TOR and two RS cells, the number of reconstructed cells plus the modelling experiments, need to be increased to 4 TOR plus 4 RS cells.

2) The observed differential expression densities of the Kv4.3 channels in RS vs TOR cells would require systematic out-side out patch recordings along the somato-dendritic domain.

3) The authors tried to block Kv4 channels with HpTx (blocks Kv4.2 and 4.3) which increased the number of APs during the initial 150 ms of a depolarizing current step in TOR cells, however, the effect was only minimal (Figure 5A/B). Since the authors argue that auxiliary subunits are important in defining the gating properties of the Kv4.3 one option would be to knock out one or the other subunit using small interference RNA. It would be great if the proposed effect of the auxiliary subunit could be further proven.

4) Can we exclude the possibility that activity states of the CCK cells may drive the expression of the auxiliary subunits? Thus, could it be that we look on the same 'class' of CCK cells of various morphologies at different 'homeostatic' plasticity states? Some CCK INs may receive a higher drive than others resulting in differential expression profiles of auxiliary subunits that 'set' their discharge patterns.

Reviewer #2:

The manuscript by Olah et al.,. is focusing on firing phenotypes of CCK-expressing interneurons in rat hippocampal CA3 area. Using a combination of whole-cell and outside-out patch-clamp recordings, patch-seq, IHC analysis and computational modelling, the authors report two firing phenotypes in the population of CCK+ cells. In particular, the low-voltage-activated delayed firing due to Kv4.3 channels and distinct Kv4 auxiliary subunits is revealed in ~50% of CCK+ cells. When put in the cell model with realistic synaptic inputs mimicking in-vivo-like conditions, this feature had impact on the cell excitability during 8-15 Hz oscillations. This work is very interesting as it demonstrates how differences in the K^+^ channel auxiliary subunit isoforms may affect the cell functional phenotype; it therefore brings a significant contribution to the field. However, additional analysis and text revision will be required to support the major conclusions. Specifically, additional efforts are required to (1) clarify on the cell types/identity, (2) expand the patch-seq analysis, and (3) discuss the functional significance of the observed phenomenon. Below I summarize these points:

1) Cell types/identity: the cells were identified as CCK+ based on post-hoc IHC immunolabelling. Only half of these cells exhibited the TOR firing, suggesting that there may be two different CCK+ cell sub-types. The authors put additional efforts to examine the morphological properties of recorded cells. 172 neurons were reconstructed and identified as BCs, MFAs and SC-ACs. While the TOR firing was detected in morphologically different CCK+ cell types, it would be important to validate this observation by performing cluster analysis combining electrophysiological and morphological features of recorded cells (see Hosp et al., 2014). The authors should including additional membrane properties: membrane time constant, slow AHP, Ih, rebound depolarization (e.g. CA1 CCK+ BCs and SC-ACs have different Ih and rebound depolarization, Evstratova et al., 2011)) and firing parameters (e.g., discharge frequency, ISIfirst/last, CV of ISI at different Vm levels etc.) in the cluster analysis and combine it with morphological features (e.g., axon distribution in different sublayers, dendritic features) to explore CCK+ CA3 population with TOR vs RS firing. The SC-ACs reconstructions are not illustrated in the manuscript, and, as the authors mention in the Materials and methods section, distinguishing BCs from MFA cells may be challenging due to their partially overlapping axonal termination zones. Quantitative analysis of the axon distribution is required to validate the cell identification.

2) Patch-seq analysis is prone to contamination as can be concluded from most of articles published so far using this technique (e.g., see Tripathy et al., 2018; for in-depth assessment of this issue). The authors should provide data on sample contamination before any conclusion can be made. Expression of excitatory, microglial, astrocytic and oligo markers should be shown next to the gene expression data. Negative control data from the rat CA3 region (see Luo et al., 2019) should be included. As far as I can see, cells included in patch-seq were only identified based on firing pattern and expression of Cck and Cnr1 genes. The authors should confirm the morphology of the included cells, as many interneuron types express Cck and Cnr1 genes (e.g., VIP+ subiculum-projecting cells, Luo et al., 2019). In relation to the previous point (cell identity), the authors should provide more information on gene expression by TOR vs RS cells: common vs specific genes for each firing phenotype. It is unclear how the cluster analysis of gene expression was performed (on genes detected in 3 out of 17 cells?). This analysis needs to be shown. While normalization for gene expression is a good approach, for comparison with previous published data it would be important to show the gene expression in TPM. What is shown in Figure 6A: mean +/- SE? I'm not sure that this is the right way to illustrate this data given a high variability in gene expression between individual samples. Here, the authors claim significant differences in the expression of Kv1.3, Kv3.2 and Kv4.3 genes but it is not indicated on the figure. Was it statistically significant?

3) Functional significance of two firing phenotypes within the same interneuron population (if the latter is confirmed after additional analysis) needs to be discussed more. Having two firing phenotypes within the same cell type would increase the overall variability in their firing, thus decreasing their reliability under some conditions. Under which in-vivo conditions it may happen? Do CCK+ cells exhibit a hyperpolarized Vm of -70 to -90 in vivo? What will be the network outcome given that apparently different cell types such as BCs, SC-ACs and MFAs can show this phenomenon?

Reviewer #3:

The paper by Olah et al., examines the molecular determinants of the action potential phenotype of CCK+ interneurons in the hippocampus. To address this question, the authors use electrophysiology, immunohistochemistry, single-cell RNA seq, and modeling. The main findings are:

- CCK+ interneurons in CA3 fall into two functional classes with distinct action potential phenotypes.

- Different excitability properties lead to differential recruitment of these interneurons during theta network oscillations.

- KV 4.3 contributes to the excitability in the two classes of cells, but does not explain the difference in the action potential phenotype.

- Firing properties are correlated with differential expression of Kv4 auxiliary subunits (KChIP1 vs. KChIP4e and DPP6S).

Based on these results, the authors conclude that alternative splicing of a small number of genes significantly changes the action potential phenotype of a neuron. Overall, I found this a quite nice paper. The question is interesting, the experiments are technically well performed, the results are mostly convincing, and the paper is clearly written. However, several points need to be addressed before the manuscript can be further considered.

Essential revisions:

1) The links between auxiliary subunit expression and channel function remain loose. If I read the statements correctly, (1) the effects of KChIP4e have not been investigated in recombinant expression systems, (2) it is unclear whether DPP6L and DPP6S shift the activation and inactivation curves in the proposed manner in recombinant channels, and (3) it is unclear whether ternary or higher order complexes (subsection “Same channel protein but distinct auxiliary subunits are responsible for the different I_SA_ currents and for the different functionality of TOR and RS cells”) show the proposed properties when recombinantly expressed. Ideally, the authors should recombinantly express the proposed subunit combinations and compare the functional properties. At the very least, they should better discuss the similarities and differences between native and recombinant channels. A systematic comparison of properties in a supplementary table might help.

2) Several conclusions are based on analysis of K^+^ currents in the whole-cell voltage-clamp configuration. However, voltage-clamp errors may be a problem in these measurements. At the very least, they should better discuss this point. Additionally, pharmacological isolation of Kv4 currents in high TEA concentrations might be useful to improve voltage clamp.

3) Activation and inactivation curves of several ion channels are known to be affected by patch configuration and time-dependent changes, e.g. shifts to more negative values over time (Donnan effects may contribute to these changes). This raises the question whether the curves were similar in different recording configurations and affected by nonstationarities. At the very least, statements should be added to the Methods section.

4) The authors conclude that fine-tuning in gating and conductance density of K^+^ channels is critically important for determining the action potential phenotype. This seems in contrast to the previous demonstration that the same firing pattern can be generated by several combinations of conductances (Marder and Goaillard, 2006). The authors should better discuss these opposing views.

5) The evidence for an involvement of alternatively spliced auxiliary subunits is compelling but remains correlative. The authors should clearly mention the limitations of the correlative approach in both Abstract and Discussion section.

6) The modeling section is unclear. Inclusion of a table with the main free parameters may provide the necessary clarification. Furthermore, the Materials and methods section addressing this part needs to be rewritten. Finally, it is not entirely clear whether two populations of channels are indeed required to describe I_SA_ channel gating in TOR cells. A single population with intermediate gating properties may do the same job.

7) Surely, mRNA and protein expression are important, but acute modifications (e.g. by phosphorylation) may be also relevant. At the very least, appropriate caveat sentences should be added.

8) Finally, the paper is too long in relation to the novel information it contains. It should be shortened to approximately 60 to 70%. A scheme of the proposed mechanisms at the end of the paper might increase clarity and help in shortening the text.

---

## [Author Response]

Essential revisions:1) Improve the cell identity following the suggestions of reviewer #2.

As requested by the reviewer new data and analysis were included. We now show cluster analysis using multiple different parameters, including electrophysiology, morphology and mRNA content (Figure 2—figure supplement 2). Furthermore, we present more detailed data about the electrophysiological and morphological properties of our CCK+IN data set (Figure 1—source data 1). For more information see our response to reviewer #2, point 1, which is a related question.

2) Perform additional controls for patch seq analysis as suggested by reviewer #2.

We have revised the RNA data and performed new analysis as requested. For details see our response to reviewer #2 (point 2) and the revised Figure 2 and Figure 5 and text subsection “TOR and RS firing types do not correlate with previously known subtypes of CCK+ cells”, Materials and methods section. Briefly, we tested contamination of our CCK+IN samples by other cells, including microglia, oligodendrocytes, astrocytes and excitatory cells. This analysis revealed that one of our cells was likely contaminated by microglia and, therefore, it was removed from the analysis. However, our original conclusions remain unchanged.

We also added new analysis using hierarchical cluster analysis of CCK+INs based on their total gene expression as well as their physiological properties (see new Figure 2—figure supplement 2). For more information see our response to reviewer #2.

Finally, we now include an additional potential marker of a subpopulation of CCK+INs, *Lypd1*, which was previously described in a subpopulation of CA1 CCK+INs (Harris et al., 2018), but is expressed in the majority of TOR cells per our RNAseq analysis.

3) Improve the discussion on the potential functional role of both interneuron types as questioned by reviewer #1 and 2.

We revised the manuscript regarding the discussion about the potential role of the two firing types of CCK+INs as suggested by the reviewer. For more information, see our response to reviewers #1 (point 4) and reviewer #2 (point 3).

4) Discuss the similarities and differences between native and recombinant channels KChIP4e. If data to recordings from recombinant channels are available, include them in the study. If they are not available, discuss the limitations of your data (see comment of reviewer #3).

We thoroughly revised the manuscript to better explain the functional diversity of KChIPs and to highlight that KChIP4e, whose presence correlates with TOR firing in CCK+INs, is the least known among the auxiliary subunits of Kv4 channels (see revised text in subsection “Auxiliary subunits of Kv4 channels in TOR and RS cells”, Discussion section).

The precise electrophysiological properties of KChIP4e containing channels are not available from isolated expression systems. There are apparent contradictions between the results of previous papers, which may likely stem from the functional diversity of different KChIP splicing variants. In fact, there are often larger differences between the effects of the different variants of the same KChIP proteins than between different KChIP proteins. For example, the effects of KChIP4a are more similar to KChIP4e, KChIP2x or KChIP3x than to KChIP4bL. By contrast, KChIP4bL shows functional similarities with KChIP1a, KChIP1c, KChIP2a, KChIP3a and KChIP4d. The first group forms the so-called tmKChIPs, which retain Kv4.3 from the plasma membrane, which we also observed in TOR cells were KChIP4e is abundant. In expression systems, most tmKChIPs slow inactivation and affect voltage dependence, but the effect of KChIP4e on the kinetics of the Kv4.3-mediated current remains elusive. The evaluation of KChIP-effects is further complicated by potential interactions with DPP proteins, because the effects of DPPs and KChIPs do not appear to be simply additive. We would like to emphasize that, although it is not possible to prove now that KChIP4e is responsible for the unique subthreshold potassium conductance and firing of TOR cells, our results show strong correlation between gene expression and function. We found that KChIP4e is exclusive to TOR cells, whereas KChIP1 is abundant in RS cells. Furthermore, the KChIP sets did not overlap between the two types. We thoroughly revised the discussion regarding KChIP and DPP functions to better reflect the current state of knowledge in this field. We also explicitly state that the precise contribution of KChIP4e is not completely understood. However, the structural similarities of KChIP4e to other tmKChIPs and their similar effects on the surface expression suggest a causal relationship between this subunit and the unique properties of I_SA_ currents in TOR cells.

5) Improve the discussion on errors related to voltage-clamp recordings of K^+^ channel-mediated currents.

To address this question, we have performed new experiments that are less affected by voltage-clamp errors. Specifically, we compared potassium currents in nucleated patches from the two types of CCK+INs in the presence of TEA, which inhibits most Kv channels, but leaves Kv4 unaffected. We also tested the effects of specific Kv4 channel inhibitor, HpTX on these currents. This experimental configuration allowed recording of well-controlled Kv4-mediated currents even with large voltage steps and the direct comparison of the properties and densities Kv4-mediated conductances in TOR and RS cells.

The new data not only address the questions of better voltage control but strengthens our conclusions. These results provide a more direct evidence for left-shifted voltage dependence and slower inactivation kinetics of the current in TOR cells. Importantly, the results also confirm that RS cells have larger Kv4.3-mediated total conductance than TOR cells, which accords with the expected effects of KChIP1 and KChIP4e. This new data is now shown in Figure 3C. For additional information, see our response to reviewer #3 point 2.

6) Activation and inactivation curves of several ion channels are affected by patch configuration and time-dependent changes, e.g. shifts to more negative values over time (Donnan effects may contribute to these changes). This raises the question whether the curves were similar in different recording configurations and affected by nonstationarities. At the very least, statements should be added to the Materials and methods section.

We agree that long-lasting hyperpolarization may cause changes in the ionic milieu. However, our experiments were devised to minimize this effect. We applied large hyperpolarization only in a few experiments and they lasted not longer than 3 seconds in the current clamp recordings. We highlight this in the manuscript (subsection “A large portion of CCK+INs show state-dependent firing in the CA3 area” and Materials and methods section).

Furthermore, we would like to emphasize that I_SA_ currents from TOR and RS cells were compared under the same conditions, including whole-cell current- and voltage-clamp, outside-out patches and nucleated patches. Recording conditions were stable in individual cells and the differences were consistent. Thus, it is unlikely that the Donnan-effect influenced the two cell types differently.

Third, now we also include detailed data about the stability of firing patterns of individual cells during long recording sessions (>45 minutes, see subsection “A large portion of CCK+INs show state-dependent firing in the CA3 area” and subsection “Kv4 channels are responsible for both types of I_SA_ currents in CCK+INs”). For more information regarding this question, see our response to reviewer #3.

7) Discus the opposing views that different K^+^ channels or fine-tuning in gating of the same K^+^ channels determines the action potential phenotype as suggested by reviewer #3.

We have included a new subsection in the Discussion section about the functional convergence of different conductance sets and the effect of a single conductance.

8) Improve the discussion on the correlative nature of evidences between alternatively spliced auxiliary subunits and their effects on neuronal activity patterns.

We have revised the Discussion section as suggested to highlight that some of our findings provide correlative evidences between the availability of different KChIPs and firing properties of TOR and RS CCK+INs.

The correlative nature of the evidences is clearly emphasized in the abstract: “the firing phenotypes were correlated with the presence of distinct isoforms of Kv4 auxiliary subunits” and “alternative splicing of few genes… can determine cell-type identity “.

9) Improve the modeling section by following the suggestions of reviewer #3.

We have included a summary table about the ionic conductances in the models as suggested by reviewer #3 (see Figure 4—figure supplement 1). Furthermore, we added more reconstructed cells to the modelling, which now includes 10 cells. Furthermore, we revised the description of the modelling as suggested. For additional information, see our response to reviewer #3 point 6.

10) Discuss, that acute modifications such as phosphorylation could also influence the gating kinetics of the channels.

As suggested by the reviewers we now discuss that differential phosphorylation of Kv4 channel complexes may also contribute to differences in the kinetic properties of the currents. We also cite relevant literature about sub-type-specific contribution of KChIPs to the different phosphorylation states of Kv4 channels as a potential molecular mechanism for the modulation the channel (Discussion section). See our response to reviewer #3.

11) Increase the number of modeled single cells following reviewer #1.

We have reconstructed an additional five cells and performed the same simulations as we did with the original five cells. The simulations on the new cells confirmed our previous observations. Furthermore, we used this additional analysis to present more detailed comparisons regarding the morphology of TOR and RS cells (Figure 2—figure supplement 1).

List of new experiments:

- Nucleated patch recordings to measure Kv4-mediated currents in the two types of CCK+INs (Figure 3C).

- Five new, completely reconstructed cell morphologies (Figure 2 and Figure 2—figure supplement 1 and Figure 2—figure supplement 2).

- All simulations were repeated with the five newly reconstructed cells (Figure 4 and its supplements).

List of new analysis:

- Additional cluster analyses considering gene expression, morphology and firing in various combinations (Figure 2—figure supplement 2A-G).

- Additional morphological analysis to map the primary axon termination zone of the ten reconstructed cells (Figure 2—figure supplement 2H).

- Dendritic distance-dependence of current density (Figure 3—figure supplement 1C).

- We re-analyzed the SC-RNAseq data and correlations after the removal of one cell from the data set based on the suggestion of reviewer #2.

New figure panels and tables:

- Figure 3C: nucleated patch recordings of Kv4.3-mediated currents from TOR and RS cells.

- Figure 2 – figure supplement 1: Scholl-analyses of 3-dimensionally reconstructed morphologies of CCK+INs.

- Figure 2—figure supplement 2: cluster analysis of CCK+INs based on active genes, electrophysiological and morphological properties.

- Figure 1—source data 1: detailed electrophysiological parameters of CCK+INs.

- Figure 4—figure supplement 1: conductance and membrane parameters of the modelled CCK+INs.

Revised figure panels:

- The original Figure 3 and Figure 5 are merged into a single figure. This new figure now summarizes all experiments regarding Kv4-mediated current and conductance parameters in CCK+INs.

- Figure 2C, D: revised data (see our response to reviewer #2 point 1).

- Figure 4D: updated with additional cells, now n = 10 cells.

- Figure 5 (previously Figure 6): revised data (see our response to reviewer #2 point 1).

Reviewer #1:[…]1) In view of the fact that a large body of modelling experiments are based on only three TOR and two RS cells, the number of reconstructed cells plus the modelling experiments, need to be increased to 4 TOR plus 4 RS cells.

We have reconstructed five more cells and included them in all simulations. The results of the five new cells were similar to those obtained in the original five cells. Revised Figure 4 now shows the average outcomes of ten simulations.

2) The observed differential expression densities of the Kv4.3 channels in RS vs TOR cells would require systematic out-side out patch recordings along the somato-dendritic domain.

These data are central to the revised manuscript (see subsection “Differences in low-voltage-activated potassium currents (I_SA_) underlie the heterogeneity of CCK+IN firing” and Figure 3—figure supplement 1). The observed current densities were similar along the somato-dendritic axis both in RS and TOR cells. In addition, we performed new experiments using the nucleated patch configuration, which allowed the direct comparison of Kv4-mediated conductances between RS and TOR cells (Figure 3C).

3) The authors tried to block Kv4 channels with HpTx (blocks Kv4.2 and 4.3) which increased the number of APs during the initial 150 ms of a depolarizing current step in TOR cells, however, the effect was only minimal (Figure 5A/B). Since the authors argue that auxiliary subunits are important in defining the gating properties of the Kv4.3 one option would be to knock out one or the other subunit using small interference RNA. It would be great if the proposed effect of the auxiliary subunit could be further proven.

Please note that we used rats for our experiments for which knock-out models are not easily accessible. However, to strengthen our conclusion for the HpTX effects on firing, we performed new experiments with nucleated patches. The results revealed that HpTX-effects are voltage-dependent, which is consistent with previous reports that we cited in the manuscript. In fact, HpTX does not simply block Kv4.3 currents, but shift the activation curve to the right, which limits the channel availability. Therefore, at near threshold voltages HpTX results in 19.3% inhibition of Kv4.3-mediated currents in TOR cells (see new Figure 3C panel). Thus, the incomplete inhibition explains our observations on the HpTX-effect on the firing.

We decided against the employment of an shRNA knock-down strategy because it would not be possible to prove whether the infected cells would have been TOR or RS phenotypes without the Kv4.3-KD and because this channel is present in both cell types.

4) Can we exclude the possibility that activity states of the CCK cells may drive the expression of the auxiliary subunits? Thus, could it be that we look on the same 'class' of CCK cells of various morphologies at different 'homeostatic' plasticity states? Some CCK INs may receive a higher drive than others resulting in differential expression profiles of auxiliary subunits that 'set' their discharge patterns.

We have revised the manuscript to raise the possibility that CCK+INs may switch between TOR and RS firing modes (see the new discussions in subsection “I_SA_ current properties tune CA3 CCK+IN function” and subsection “Same channel protein, but distinct auxiliary subunits may be responsible for different I_SA_ currents in TOR and RS cells). To address this point experimentally, we have tried to alter general signaling mechanisms by changing the extracellular or intracellular calcium ion concentration and assess if these were sufficient to change the RS or TOR firing type. However, none of these trials suggested switch between the two firing phenotypes.

We also state in the Results section that based on the rate and amplitude of spontaneous events, the excitatory drive to TOR and RS cells appears to be similar (subsection “A large portion of CCK+INs show state-dependent firing in the CA3 area”).

Reviewer #2:[…] However, additional analysis and text revision will be required to support the major conclusions. Specifically, additional efforts are required to (1) clarify on the cell types/identity, (2) expand the patch-seq analysis, and (3) discuss the functional significance of the observed phenomenon. Below I summarize these points:1) Cell types/identity: the cells were identified as CCK+ based on post-hoc IHC immunolabelling. Only half of these cells exhibited the TOR firing, suggesting that there may be two different CCK+ cell sub-types. The authors put additional efforts to examine the morphological properties of recorded cells. 172 neurons were reconstructed and identified as BCs, MFAs and SC-ACs. While the TOR firing was detected in morphologically different CCK+ cell types, it would be important to validate this observation by performing cluster analysis combining electrophysiological and morphological features of recorded cells (see Hosp et al., 2014). The authors should including additional membrane properties: membrane time constant, slow AHP, Ih, rebound depolarization (e.g. CA1 CCK+ BCs and SC-ACs have different Ih and rebound depolarization, Evstratova et al., 2011)) and firing parameters (e.g., discharge frequency, ISIfirst/last, CV of ISI at different Vm levels etc.) in the cluster analysis and combine it with morphological features (e.g., axon distribution in different sublayers, dendritic features) to explore CCK+ CA3 population with TOR vs RS firing. The SC-ACs reconstructions are not illustrated in the manuscript, and, as the authors mention in the Methods section, distinguishing BCs from MFA cells may be challenging due to their partially overlapping axonal termination zones. Quantitative analysis of the axon distribution is required to validate the cell identification.

We have added detailed data about the electrophysiological parameters of all recorded cells in Figure 1—source data 1, and detailed morphological data of ten reconstructed cells (Figure 2—figure supplement 1). Note that we added five new morphologies in the revised manuscript.

The quantitative comparison of the electrophysiological and morphological parameters, together with the actively transcribed genes are shown in Figure 2—figure supplement 2 (see also subsection “TOR and RS firing types do not correlate with previously known subtypes of CCK+ cells”). Cluster analysis was not able to distinguish TOR and RS cells based on their morphological features or gene expression. However, as a positive finding, cluster analysis separated out cells based on known CCK+IN morphological subcategories, such as basket- and two types of dendrite targeting cell types (Figure 2—figure supplement 2H).

The reconstructed cells represent different morphological subtypes, which can be classified based on the target zone preference of their axons as basket, mossy fiber associated and dendrite-targeting perforant path-associated cells. Our initial hypothesis, similar to what this Reviewer suggests, was that the two types of firing correlate with known subgroups of CCK+INs. However, our results prompt us to reject this hypothesis.

2) Patch-seq analysis is prone to contamination as can be concluded from most of articles published so far using this technique (e.g., see Tripathy et al., 2018; for in-depth assessment of this issue). The authors should provide data on sample contamination before any conclusion can be made. Expression of excitatory, microglial, astrocytic and oligo markers should be shown next to the gene expression data. Negative control data from the rat CA3 region (see Luo et al., 2019) should be included. As far as I can see, cells included in patch-seq were only identified based on firing pattern and expression of Cck and Cnr1 genes. The authors should confirm the morphology of the included cells, as many interneuron types express Cck and Cnr1 genes (e.g., VIP+ subiculum-projecting cells, Luo et al., 2019). In relation to the previous point (cell identity), the authors should provide more information on gene expression by TOR vs RS cells: common vs specific genes for each firing phenotype. It is unclear how the cluster analysis of gene expression was performed (on genes detected in 3 out of 17 cells?). This analysis needs to be shown. While normalization for gene expression is a good approach, for comparison with previous published data it would be important to show the gene expression in TPM. What is shown in Figure 6A: mean +/- SE? I'm not sure that this is the right way to illustrate this data given a high variability in gene expression between individual samples. Here, the authors claim significant differences in the expression of Kv1.3, Kv3.2 and Kv4.3 genes but it is not indicated on the figure. Was it statistically significant?

We thank the reviewer for raising this excellent point. While off-cell contamination is a concern, our previous study did not indicate off-cell mRNA as a major source of contamination (Lukacsovich et al., 2019). Furthermore, we found that some of the genes that we previously regarded as contamination may be cell type specifically expressed in select interneuron types (Winterer et al., 2019). We therefore did not include negative or “empty” controls in this study. Because such controls should be properly run alongside the single-cell samples, we also decided not to generate a new control data set on its own. However, we carefully considered the reviewer’s suggestion as follows.

We assembled lists of genes which would be considered microglia- (Tyrobp, Ctss, C1qc, Cyba, Ly86), astrocyte- (Gfap, Aldh1l1, Sox9, S100b, Slc1a2), oligodendrocyte- (Olig1, Fgf2, Mtcp1, Olig2, Olig3) and pyramidal cellspecific (Baiap2l2, Slc17a7, Ptk2b, Nrn1, Fhl2, Itpka, Neurod6, Nptx1, Sv2b, Kcnv1), and tested these against our single-cell samples. We found that one of our cells showed significant expression of all 5 microglia-related genes and based on this we decided to completely exclude this cell from the study. We now specifically mention this in the revised manuscript (Materials and methods section). Testing of the other 16 cells did not reveal gene expression patterns characteristic to the above cell types.

For transcriptomic analysis of CCK+ cells, we used the inclusion criteria of the cells expressing both Cck and Cnr1 (as described in subsection “TOR and RS firing types do not correlate with previously known subtypes of CCK+ cells”). Furthermore, we would like to emphasize that all cells showed typical CCK+INs firing patterns at slightly depolarized membrane potentials (between -60 and -65 mV), such as accommodation of firing, and moderate AP width (see our response above). We agree with the reviewer that expression of these two genes does not unambiguously identify the recorded cells as either SCA, MFA or BC. However, our goal was to include diverse populations of the large CCK+IN group, as we showed in Figure 1, that TOR and RS firing types are present in all morphological classes. Unfortunately, in the RNAseq analyzed cells, labeling was incomplete in most cases (13 out of 16). Nevertheless, at least one basket, mossy-fiber-associated and Schaffer-collateral associated cells were included in these 16 cells which were identified based on partial axonal recovery. We now state this in the manuscript (Materials and methods section). Since we already demonstrated that TOR and RS firing cells are present among all 3 morphological populations, we do not expect that any of our conclusions would change even if we possessed morphological reconstruction from each of the transcriptomically analyzed cells.

We also made further clarifications about the SC-RNAseq data in the manuscript (see subsection “TOR and RS firing types do not correlate with previously known subtypes of CCK+ cells”).

We now show cluster analysis of cells in a new supplementary data set based on their firing pattern and RNA content or morphological characteristics. We would like to emphasize that the cluster analysis was done based on genes that “were detected in at least 3 of the 16 tested cells”. We used 3 as an arbitrarily determined cut-off, which represented at least 35% of the cells in TOR group (n=8 and 8; Figure 2). Using this cut-off, we aimed to not use genes which have very sparse expression in either population, but also not exclude genes, which may be sparsely but specifically expressed in one of the groups.

For comparability with other data sets, we now include TMP values in the online data repository together with the normalized data.

Significant differences in Kv genes are now highlighted in the revised Figure 5 as suggested.

3) Functional significance of two firing phenotypes within the same interneuron population (if the latter is confirmed after additional analysis) needs to be discussed more. Having two firing phenotypes within the same cell type would increase the overall variability in their firing, thus decreasing their reliability under some conditions. Under which in-vivo conditions it may happen? Do CCK+ cells exhibit a hyperpolarized Vm of -70 to -90 in vivo? What will be the network outcome given that apparently different cell types such as BCs, SC-ACs and MFAs can show this phenomenon?

Thank you for these suggestions. We revised the manuscript text accordingly. We discuss in more detail the potential functional implications of the two types of CCK+INs during different brain states, synaptic networks, and reworked the parts regarding the interpretation of findings within the Discussion section.

We now also explicitly state in the Results section that the membrane potential remained in physiological ranges during oscillating inputs (see subsection “TOR cells are selectively silenced by I_SA_TOR in a narrow range of oscillatory states”).

We also better describe that our simulations are aimed to address the question whether the two distinct firing modes of CCK+INs favor distinct in vivo oscillations and conditions (subsection “Realistic models of TOR and RS firing”, subsection “TOR cells are selectively silenced by I_SA_TOR in a narrow range of oscillatory states”). With the aggregated inputs, our aim was to generate physiologically relevant membrane potential oscillations that are driven by realistic synaptic events. Synaptic inputs, ionic conductances and cellular morphologies were virtually identical in RS and TOR simulations. Properties of the subthreshold-activated potassium conductances were the only varied parameter, which was also the subject of our study.

Reviewer #3:[…]Essential revisions:1) The links between auxiliary subunit expression and channel function remain loose. If I read the statements correctly, (1) the effects of KChIP4e have not been investigated in recombinant expression systems, (2) it is unclear whether DPP6L and DPP6S shift the activation and inactivation curves in the proposed manner in recombinant channels, and (3) it is unclear whether ternary or higher order complexes (subsection “Same channel protein but distinct auxiliary subunits are responsible for the different I_SA_ currents and for the different functionality of TOR and RS cells”) show the proposed properties when recombinantly expressed. Ideally, the authors should recombinantly express the proposed subunit combinations and compare the functional properties. At the very least, they should better discuss the similarities and differences between native and recombinant channels. A systematic comparison of properties in a supplementary table might help.

We extensively revised the manuscript to better explain the functional diversity of KChIPs and to highlight that KChIP4e is the least known among the auxiliary subunits of Kv4 channels (see below the list of corrections).

The apparent contradictions and lack of complete understanding are likely to stem from the unusually large functional diversity of different splicing variants of KChIPs. Known differences between variants of the same proteins are often larger than between different proteins. For example, the effects of KChIP4a are more similar to KChIP2x or KChIP3x than KChIP4bL. By contrast, KChIP4bL shows functional similarities to KChIP1a, KChIP1c, KChIP2a, KChIP3a and KChIP4d.

The available publications on KChIP4e in expression systems focused primarily on its roles in the regulation of the surface expression of Kv4s. These results are in full agreement with our observations showing that in TOR cells where KChIP4e isoform is abundant, the amount of Kv4.3 channels in the plasma membrane is lower than in RS cells, which express other KChIP isoforms. Unfortunately, the effects of KChIP4e on the Kv4.3-mediated current kinetics are not explored. Nevertheless, KChIP4e is expected to have similar general effects on the channel kinetics as other tmKChIPs. The known tmKChIPs always slow inactivation of Kv4-madiated currents, they retain Kv4 from the plasma membrane and slow recovery from inactivation. Most of the left shifts the voltage dependence of the activation and steady-state inactivation depending on the presence of DPPs.

Our data confirmed a correlation between the presence of KChIP4e and the lower surface expression of Kv4.3, the slower inactivation and recovery kinetics, and the left shifted voltage dependence and in TOR cells. Furthermore, our new experiments with nucleated patch recordings provide now an additional direct evidence for lower Kv4.3 density in the plasma membrane of TOR cells. We hope that our correlational observation will drive future research to directly compare the combined effects of KChIP4e-DPP6S/L with KChIP1-DPP6L in because now our results provide the functional relevance for these experiments.

We thoroughly revised the manuscript at several points. For example, in the Discussion section we added the following text: “KChIP proteins and their splice-variants show unusual functional diversity. Different splicing of the same protein can have opposing effects on Kv4 functions, whereas, splice-variants of different proteins can have analogous effects. One outstanding group is the tmKChIP family that consist of KChIP2x, KChIP3x, KChIP4a and KChIP4e. Their common structural feature is an extra N-terminal hydrophobic domain that binds them to the membrane. In contrast to classical KChIPs, tmKChIPs typically retain Kv4 channels from the plasma membrane, slow the inactivation kinetics and the recovery from inactivation.”

In subsection “Same channel protein, but distinct auxiliary subunits may be responsible for different I_SA_ currents in TOR and RS cells” we also explicitly state that “The exact effects of KChIP4e on the kinetics of Kv4.3-mediated currents are not known.”

In subsection “Same channel protein, but distinct auxiliary subunits may be responsible for different I_SA_ currents in TOR and RS cells” we added: “The effects of KChIPs are not isolated from the other auxiliary subunits of Kv4. The various stoichiometries of individual Kv4 channels with DPPs and KChIPs allow delicate settings of the channel kinetics. The net effects of KChIPs and DPLPS are not simply the sum of the effects of individual subunits, and the combinatorial possibilities are not yet fully explored with the known 17 variants of KChIPs and 8 variants of DPPs.” We also modified the text accordingly in subsection “Same channel protein, but distinct auxiliary subunits may be responsible for different I_SA_ currents in TOR and RS cells”. In addition to the direct subsection “Auxiliary subunits of Kv4 channels in TOR and RS cells”.

2) Several conclusions are based on analysis of K^+^ currents in the whole-cell voltage-clamp configuration. However, voltage-clamp errors may be a problem in these measurements. At the very least, they should better discuss this point. Additionally, pharmacological isolation of Kv4 currents in high TEA concentrations might be useful to improve voltage clamp.

Thank you for raising this point. We performed new experiment to address the potential issues of whole-cell voltage-clamp recordings.

We also worried about the limited voltage clamping of the enormous I_SA_ currents. As a consequence of precaution, in the original whole-cell experiment we restricted recordings to lower voltage ranges, where the total currents are smaller.

Now, we performed new experiments using nucleated patch configuration from identified TOR and RS cells, which allows better voltage control. As suggested these experiments were made in the presence of high concentration of TEA, which does not affect Kv4-mediated currents, to further improve the voltage clamp. Furthermore, in some of these recordings we applied HpTX to further prove that I_SA_ is mediated by Kv4 channels in both types of CCK+INs. An additional advantage of this recording configuration is that it allowed recordings of Kv4-mediated currents in large voltage ranges, which is necessary to direct comparison of the amount of I_SA_ conductance in the two types of cells. The results confirmed the suggestions of the original experiments. Namely, RS cells have larger Kv4-mediated conductance than TOR cells, but the activation of I_SA_ is left-shifted in TOR cells. This confirms that, in spite of the smaller total conductance, I_SA_ in TOR cells can effectively inhibit firing. Furthermore, our data confirms previous observations about the unusual voltage dependence of HpTX blockade of Kv4, which also explains the partial inhibition of firing. The new data is shown on the revised Figure 3C and the details are described in subsection “Differences in low-voltage-activated potassium currents (I_SA_) underlie the heterogeneity of CCK+IN firing” and subsection “Kv4 channels are responsible for both types of I_SA_ currents in CCK+INs”.

3) Activation and inactivation curves of several ion channels are known to be affected by patch configuration and time-dependent changes, e.g. shifts to more negative values over time (Donnan effects may contribute to these changes). This raises the question whether the curves were similar in different recording configurations and affected by nonstationarities. At the very least, statements should be added to the Materials and methods section.

We agree that long-lasting hyperpolarization may cause changes in the ionic milieu. However, we devised our experiments to avoid this effect. We applied large hyperpolarization only in a few current experiments and in these cases, it lasted only for 3 seconds. We highlight this aspect in the manuscript (subsection “A large portion of CCK+INs show state-dependent firing in the CA3 area” and Materials and methods section). Furthermore, in voltage experiments we did not observe substantial reduction in I_SA_ currents due to shifts caused by hyperpolarization below the potassium reversal (see most left points on the inactivation curves Figure 3B, F and Figure 3—figure supplement 2C). We avoided larger hyperpolarization than -100 mV in current clamp experiments. As stated in the Materials and methods section, we typically quantified the TOR phenomenon at membrane potentials that are above the reversal potentials of potassium, which does not artificially shift potassium concentrations inside or outside the cells. The preceding hyperpolarization or depolarization lasted for 3 seconds in both RS and TOR cells. Between the traces, the membrane potential was kept at rest. Thus, depletion of potassium ions (below -90mV, the potassium reversal) was probably negligible and affected all cells similarly.

The two distinct firing patterns appeared in physiologically plausible voltage ranges, which required about only a hundred pA of hyperpolarization or depolarization from the resting membrane potential (subsection “A large portion of CCK+INs show state-dependent firing in the CA3 area”). Furthermore, not only the presence of the TOR phenomenon was stable, but the timing of the first APs was also maintained similarly in individual cells during long (>40 minutes) recordings. We now include additional experiments where we specifically investigated the stability of the firing patterns of TOR and RS cells during an hour recording period (45-64 minutes, mean: 58 min, n = 7 cells, see new text in subsection “A large portion of CCK+INs show state-dependent firing in the CA3 area”). Furthermore, we would like to point to the experiments where we investigated the effects of Kv4-inhibitor, HpTX on the firing of both TOR and RS cells (Figure 4A). In these recordings, the gap in the TOR firing recovered to control levels after the washout of the toxin. These recordings lasted for 25-58 minutes. Whereas, in RS cells the firing was stable throughout the experiments including control, toxin and washout periods, which altogether lasted for 20-58 minutes. Similarly, firing patterns remained stable during the application of TEA or low concentrations of 4-AP (Figure 3—figure supplement 1D) that do not affect Kv4-mediated potassium currents. These recordings also lasted typically for 30-50 minutes. We mention these observations in subsection “Kv4 channels are responsible for both types of I_SA_ currents in CCK+INs”.

We would like to emphasize that I_SA_ currents from TOR and RS cells were compared under the same conditions, including whole-cell current- and voltage-clamp, outside-out patches and nucleated patches. The differences were consistent, and the properties of individual cells were stable. Thus, it is unlikely that the recording configuration directly influenced these two types of firing. Our simulations also confirm that the observed differences in the I_SA_ currents alone are sufficient for generating TOR and RS firing phenotype.

4) The authors conclude that fine-tuning in gating and conductance density of K^+^ channels is critically important for determining the action potential phenotype. This seems in contrast to the previous demonstration that the same firing pattern can be generated by several combinations of conductances (Marder and Goaillard, 2006). The authors should better discuss these opposing views.

Thank you for this suggestion. We have placed our findings into the suggested context. See Discussion section.

5) The evidence for an involvement of alternatively spliced auxiliary subunits is compelling, but remains correlative. The authors should clearly mention the limitations of the correlative approach in both Abstract and Discussion section.

We have revised the Discussion section as suggested to highlight that our findings provide correlative evidence between the availability of different KChIPs and firing properties of TOR and RS CCK+INs. The correlative nature of the evidences is also clearly emphasized in the abstract: “the firing phenotypes were correlated with the presence of distinct isoforms of Kv4 auxiliary subunits” and “alternative splicing of few genes…can underlie distinct cell-type identity”.

We added the following text to the Discussion section to clarify this question:

“The presence of KChIP4e in TOR cells and KChIP1 in RS cells is consistent with their different I_SA_ kinetics and densities that underlie the different functionality of these cells. Future studies are expected not only to confirm that these alternatively spliced variants are solely responsible for the two firing types, but can also address whether these differences in subunits are determined by the destiny of the cells from early of their development or whether these subunits are actively regulated throughout the life span and may underlie activity-dependent modification of the CCK+INs population.”

Furthermore, this question is also mentioned Materials and methods section:

“…, all differences that we observed between RS and TOR cells (i.e., the higher channel density, faster inactivation kinetics and faster recovery from inactivation of I_SA_ in RS compared to TOR cells) are consistent with the differential expression of KChIP1 and KChIP4e subunits.”

We also modified sentence Discussion section which now reads as:

“We found that a potential explanation for this paradox is the differential expression of auxiliary subunits of Kv4.3 channels”.

6) The modeling section is unclear. Inclusion of a table with the main free parameters may provide the necessary clarification. Furthermore, the Methods section addressing this part needs to be rewritten. Finally, it is not entirely clear whether two populations of channels are indeed required to describe I_SA_ channel gating in TOR cells. A single population with intermediate gating properties may do the same job.

We have added a new supplementary table that summarizes the passive membrane properties and densities of active conductences in different subcellular compartments of the simulated CCK+INs (Figure 4—figure supplement 1). Here we specify which parameters were (1) variable among the ten simulated cells, (2) fixed or were responsible for the TOR and RS phenotypes. Please note that each cell was simulated both as TOR and RS phenotypes by only changing these gTOR and gRS conductances. We also added further description to the Materials and methods section to clarify these aspects.

We included both gTOR and gRS because outside out patch recordings from TOR cells showed large variability in the kinetics of the currents. Furthermore, preliminary results showed that simulations with the combined gTOR+gRS better reproduced the recorded firing patterns. We specify this in the manuscript text (subsection “Realistic models of TOR and RS firing”).

7) Surely, mRNA and protein expression are important, but acute modifications (e.g. by phosphorylation) may be also relevant. At the very least, appropriate caveat sentences should be added.

Thank you for this suggestion. We added the following text to the Discussion section to highlight posttranslational modification of Kv4 protein complexes: “In addition to the direct modulation, Kv4 protein complexes can be phosphorylated by various kinases and are involved in complex post-phosphorylation signaling, which require the presence of auxiliary subunits and modify the mediated currents. Thus, in spite of the large number of potential mechanisms that can modulate Kv4.3 functions.”

8) Finally, the paper is too long in relation to the novel information it contains. It should be shortened to approximately 60 to 70%. A scheme of the proposed mechanisms at the end of the paper might increase clarity and help in shortening the text.

Reorganization of the manuscript allowed shortening the main text. However, due to addition of new sections the overall length did not change significantly.

Because we used several approaches to unequivocally understand the underlying mechanisms the manuscript is a little bit longer than usual papers. However, without the details of the methods the results cannot be interpreted. Nevertheless, we tried to shorten and make the manuscript more concise. To facilitate this effort, majority of the requested experiments were placed into the supplementary materials and the introduction was shortened by more than hundred words. We combined Figure 3 and Figure 5 of the original manuscript into a single figure, which allowed the shortening the text and reduced the number of main figures to five.